# Systematic detection of functional proteoform groups from bottom-up proteomic datasets

Isabell Bludau [1,2], Max Frank[1,3,4,5], Christian Dörig[1], Yujia Cai [3,4], Moritz Heusel[1,6], George Rosenberger [1,7], Paola Picotti[1], Ben C. Collins [1,8], Hannes Röst [3,4 ✉] & Ruedi Aebersold [1,9 ✉]

To a large extent functional diversity in cells is achieved by the expansion of molecular complexity beyond that of the coding genome. Various processes create multiple distinct but related proteins per coding gene – so-called proteoforms – that expand the functional capacity of a cell. Evaluating proteoforms from classical bottom-up proteomics datasets, where peptides instead of intact proteoforms are measured, has remained difficult. Here we present COPF, a tool for COrrelation-based functional ProteoForm assessment in bottom-up proteomics data. It leverages the concept of peptide correlation analysis to systematically assign peptides to co-varying proteoform groups. We show applications of COPF to protein complex co-fractionation data as well as to more typical protein abundance vs. sample data matrices, demonstrating the systematic detection of assembly- and tissue-specific proteoform groups, respectively, in either dataset. We envision that the presented approach lays the foundation for a systematic assessment of proteoforms and their functional implications directly from bottom-up proteomic datasets.

[1] Department of Biology, Institute of Molecular Systems Biology, ETH Zurich, Zurich, Switzerland. [2] Department of Proteomics and Signal Transduction, Max Planck Institute of Biochemistry, Martinsried, Germany. [3] Department of Molecular Genetics, University of Toronto, Toronto, Canada. [4] The Donnelly Centre for Cellular and Biomolecular Research, University of Toronto, Toronto, Canada. [5] European Molecular Biology Laboratory, Genome Biology Unit, Heidelberg, Germany. [6] Division of Infection Medicine (BMC), Department of Clinical Sciences, Lund University, Lund, Sweden. [7] Columbia University, New York, NY, USA. [8] School of Biological Sciences, Queen's University Belfast, Belfast, UK. [9] Faculty of Science, University of Zurich, Zurich, Switzerland. ✉email: hannes.rost@utoronto.ca; aebersold@imsb.biol.ethz.ch

Human cells are known to perform thousands of different biochemical functions and the central dogma of biology states that proteins that catalyze the vast majority of these functions arise from the transcription and translation of the information contained in the respective genome. The International Human Genome Sequencing Consortium reported ~20,000 protein-coding genes in the human genome[1] and, surprisingly, the number of protein-coding genes does not scale with the complexity of functions of eukaryotic organisms[2]. These findings have led to the notion that the protein-coding information of the genome is substantially diversified structurally and functionally along the axis of gene expression[3]. Specific mechanisms that catalyze this diversification include alterative splicing of transcripts, posttranslational processing and modification of proteins, and the variable association of proteins in functional protein complexes. Consequently, protein-coding genes frequently give rise to multiple distinct protein species—proteoforms—which have a unique primary amino acid (AA) sequence and localized posttranslational modifications (PTMs)[4,5] and which might, in turn, partition into different protein complexes or show functional differences. Currently, it is estimated that the ~20,000 coding genes generate more than a million different proteoforms[6] that can differ between individual cells, tissues, and disease phenotypes[4,7–9]. This increase in complexity beyond the directly translated genomic sequence information hampers genotype-based phenotype inference and highlights the importance of capturing proteome diversity to increase the mechanistic understanding of biochemical processes in basic and translational research.

Over the last decades, mass spectrometry (MS) has emerged as the key technology for proteomic analyses[10,11]. The large array of mass spectrometric techniques can be grouped into two main approaches: top-down and bottom-up proteomics. In top-down workflows, samples containing intact proteins are chromatographically separated, ionized, and analyzed in a mass spectrometer. Recorded spectra of both the intact and fragmented proteins determine the unique primary protein sequence and PTMs of individual proteoforms[12]. Recent top-down proteomic studies reported the identification of more than 3000 unique proteoforms originating from up to ~1000 individual genes[13,14]. Gaining deeper proteoform coverage by top-down proteomics is challenged by the limitations of current separation techniques, the MS and tandem MS (MS/MS) analysis of large ions, and the interpretation of the resulting spectra by available analysis software[12,15]. Although top-down proteomics provides unprecedented insights into proteoform diversity and some proteoforms have successfully been annotated with molecular functions and implicated phenotypic traits[7], the systematic assessment of proteoform-specific functions remains challenging. A shift of focus from the mere enumeration of various proteoforms detected from a cell towards establishing direct links between proteoform species and their functional significance would be a major advance in the field.

Bottom-up proteomics is the more widely used technique for proteome-wide studies, because some of the technical challenges facing top-down proteomics are alleviated. Here, proteins are enzymatically digested into smaller peptide sequences, which are subsequently separated by liquid chromatography, ionized and analyzed by MS/MS. The identity and quantity of proteins in the tested sample are subsequently inferred from the peptides that are identified based on the acquired precursor and fragment ion spectra. The method is technically robust and has demonstrated the detection of translation products of the vast majority of coding genes in a number of species. However, bottom-up proteomic workflows suffer from the principal limitation that the connectivity between identified peptides and their proteins of origin is lost during the enzymatic digestion step. This necessitates an in silico inference step that maps measured peptide signals back to individual proteins. This is a challenging task in general[16] and is particularly hard for resolving different proteoforms[3]. Recent advances in instrumentation, data acquisition and data analysis, especially the development of data-independent acquisition (DIA/Sequential Window Acquisition of all Theoretical (SWATH)-MS) strategies, have enabled the measurement of large bottom-up proteomic datasets at high proteome coverage, combined with consistent and accurate quantification[17–19]. Based on these developments, the peptide-level bottom-up proteomic data became more reliable, both on the qualitative and quantitative level, as demonstrated in several of our previous studies[20,21]. Thus, useful information about the presence of individual modifications or sequence variants on the peptide level can be readily obtained. However, the possibilities to systematically assign and distinguish unique proteoforms from bottom-up proteomics datasets remains a mostly unexplored area to date.

Nevertheless, researchers in the early days of bottom-up proteomics already observed that peptides of the same protein might follow distinct quantitative patterns across a dataset, and that peptide co-variation analysis can be leveraged to improve proteomic analyses on different levels. The predominant focus of previous work has been to use peptide correlation analysis for the purpose of filtering out dissimilarly behaving peptides in an effort to improve protein quantification[22,23] or protein inference[24]. It has also been recognized that some of the determined "outlier" peptides could indeed contain valuable biological information, e.g., by originating from different proteoforms and previous work explored the possibility to use peptide correlation patterns for proteoform assignment[23,25–27].

In this work, we present COPF, a strategy for COrrelation-based functional ProteoForm assessment in bottom-up proteomics data. COPF extends the concept of peptide correlation analysis towards establishing a generic workflow with the main purpose of systematically assigning peptides to co-varying proteoform groups (also see Glossary in Supplementary Table 1). We benchmark COPF against PeCorA, a state-of-the-art tool for proteoform identification in bottom-up proteomics data[27], demonstrating that COPF performs better in the detection of proteoforms differing by multiple peptides. Furthermore, our data show that COPF results are based on a conservative and well-calibrated error model, and that the strategy is applicable to complex experimental designs and also the analysis of a single condition. We first demonstrate the capabilities of COPF by applying it to a dataset where cells in two cell cycle stages are compared. The dataset was generated by protein complex co-fractionation via size-exclusion chromatography (SEC) coupled to DIA/SWATH-MS[28]. The results indicate that COPF is capable to systematically detect assembly- and cell cycle-specific proteoform groups. As a second example, we apply COPF to assign functional proteoform groups in a typical bottom-up proteomic cohort study consisting of five tissue samples from the mouse BXD genetic reference panel[29]. In this dataset, COPF could determine several tissue-specific proteoform groups. The wealth of biological information that COPF provides for both the cell cycle SEC-SWATH-MS and mouse tissue datasets can be further investigated on the online platform that we provide for manual data exploration: http://proteoformviewer.ethz.ch/. The COPF algorithm is fully integrated and is available within the CCprofiler framework[21,30]. It includes specific modules to assess the biological credibility of detected proteoform groups and the unique possibility to directly integrate COPF results into protein complex analysis to determine assembly-specific proteoforms. We envision that COPF can make a significant contribution towards the

systematic assessment of proteoform groups across large bottom-up proteomic datasets and for linking these groups to biological functions.

## Results

**Principle of the method and implementation.** The assignment of peptides to unique proteoforms is a challenging task in bottom-up proteomic workflows, because the majority of detected peptides are frequently shared between multiple proteoforms and multiple diverging peptides cannot be uniquely assigned during protein inference. Here we propose a data-driven strategy to assign peptides to unique functional proteoform groups based on peptide correlation patterns across large bottom-up proteomic datasets (COPF). We define a functional proteoform group as a group of peptides derived from the same gene that co-vary across a large and heterogeneous dataset. A proteoform group can, but does not have to represent a unique, specific proteoform (also see Glossary in Supplementary Table 1). The COPF strategy is based on the following considerations: (i) in case only one proteoform is expressed or all proteoforms of a protein have similar characteristics across a heterogeneous dataset, all sibling peptides (i.e., peptides originating from the same parental gene/protein) should display a similar quantitative profile, as schematically illustrated in Fig. 1A left panel; (ii) if a gene generates multiple distinct proteoforms that differ between the analyzed samples across a heterogeneous dataset, sibling peptides of that protein can be separated into groups of highly correlated peptides, which we accordingly assign to distinct proteoform groups, schematically illustrated in Fig. 1A right panel.

Conceptually, the proteoform detection workflow in COPF can be divided into four steps (Fig. 1A, computational data analysis). First, the intensities of peptides assigned to the same gene or protein identifier are determined from the corresponding MS signals across all measured samples. Second, all pairwise peptide correlations within a protein are calculated based on the determined intensity values across samples. Third, the peptides of a protein are subjected to hierarchical clustering, using one minus the previously calculated correlation as the distance metric. The tree is then cut into two clusters, minimally containing two peptides each (see "Methods" section for details). Fourth, a proteoform score is calculated for each protein. The score is calculated as the mean peptide correlation across clusters minus the within-cluster correlation. A higher proteoform score thus indicates a higher within-cluster vs. across-cluster correlation. The assumption of our COPF strategy is that proteins with multiple distinct proteoforms that behave differentially across a dataset will have higher proteoform scores than proteins without differentially behaving proteoforms. Finally, COPF estimates $p$-values for each proteoform score and performs multiple-testing correction (see "Methods" section for details).

It is important to highlight that COPF analysis does not require any prior definition of biological conditions or a specific experimental design, as it exploits inherent variation in the data independent of its origin. Thus, COPF has the unique benefit that it is applicable to non-pairwise comparisons or to comparisons that do not include multiple conditions, exemplified by continuous data such as a single SEC-SWATH-MS experiment where only one condition is analyzed. In addition, COPF can also be applied to data with complex, nested designs including multiple covariates. In fact, the correlation-based approach employed by COPF is particularly powerful when applied to large and heterogeneous datasets, even in the absence of an explicit reference condition. These are typically difficult to assess by other approaches, such as PeCorA[27], which require a homogeneous reference condition.

The COPF strategy is implemented and openly available as an extended version of our previously published CCprofiler R package[21,30,31] at: https://github.com/CCprofiler/CCprofiler. The CCprofiler framework offers the unique possibility to directly integrate COPF results into the analysis of protein complex assemblies by SEC-SWATH-MS or similar co-fractionation MS approaches (Fig. 1B). Proteoform groups assigned by COPF can thereby directly be classified as assembly-specific and/or as condition-specific if multiple conditions are analyzed. To gain even more insights into the characteristics of the detected proteoform groups, we further implemented a peptide proximity analysis as a post-processing module in COPF (Fig. 1C). The proximity analysis evaluates whether peptides assigned to the same proteoform group are in closer relative sequence proximity than expected for random peptide grouping. This module identifies cases where proteoforms differ by extended sequence stretches, e.g., when generated by alternative splicing or proteolytic cleavage. The post-processing modules of COPF that are available within the CCprofiler framework therefore provide the opportunity to not only detect proteoform groups but to also systematically assess some of their biological characteristics.

**Benchmark.** A particular challenge for benchmarking the COPF strategy for assigning functional proteoform groups is the lack of available biological ground truth data. To evaluate COPF performance, we therefore conducted a sensitivity analysis based on an in silico generated benchmarking dataset and compared the COPF results with those from PeCorA, a recently published tool for proteoform assessment[27]. We used a subset of the SWATH-MS multi-laboratory study[19] as the basis to generate the benchmarking data. The selected dataset contains 21 replicate MS analyses of HEK293 cell lysates measured on 3 days over a week (days 1, 3, and 5, respectively). To introduce quantitative variation, we generated in silico fold-changes by adjusting intensities of day 3 and day 5 data with two randomly selected factors between 1 and 6. Subsequently, artificial proteoforms were introduced for 1000 proteins by selecting a specified number of peptides for which the intensity values in the day 5 data were adjusted by a random factor between 0.01 and 0.9. Example intensity profiles for a protein with two proteoforms (top) and a protein with only a single proteoform (bottom) are shown in Fig. 2A. The resulting set of proteins with either a single or two proteoforms was then used to evaluate the ability of our algorithm to correctly distinguish proteins with a single or multiple proteoforms, as well as to assign peptides correctly to their corresponding proteoform group. Figure 2B shows the pseudo-volcano plot for proteins with two (green) and single (orange) proteoforms. Here, proteoforms were generated by randomly perturbing between 2 peptides and 50% of the peptides in a protein. To set COPF performance metrics in context to previous work, we also analyzed our benchmarking dataset with the recently published PeCorA software[27]. For this comparison, we generated three benchmarking sets: the first consisted of proteins where proteoforms differed by a single peptide, the second of proteins where proteoforms differed by two peptides, and the third consisted of proteins where proteoforms differed by 50% of a proteins' peptides. We compared the receiver operating characteristic (ROC) curves for COPF and PeCorA for all three benchmarking sets (Fig. 2C). Here, individual points are derived by filtering the data at a specific adjusted (adj.) $p$-value threshold and showing the corresponding true positive rate (TPR) and false positive rate (FPR). COPF requires minimally two peptides to differentiate proteoform groups. As expected, COPF could not detect the proteoforms differing by a single peptide in the first benchmarking set (Fig. 2C left panel). In contrast, PeCorA was

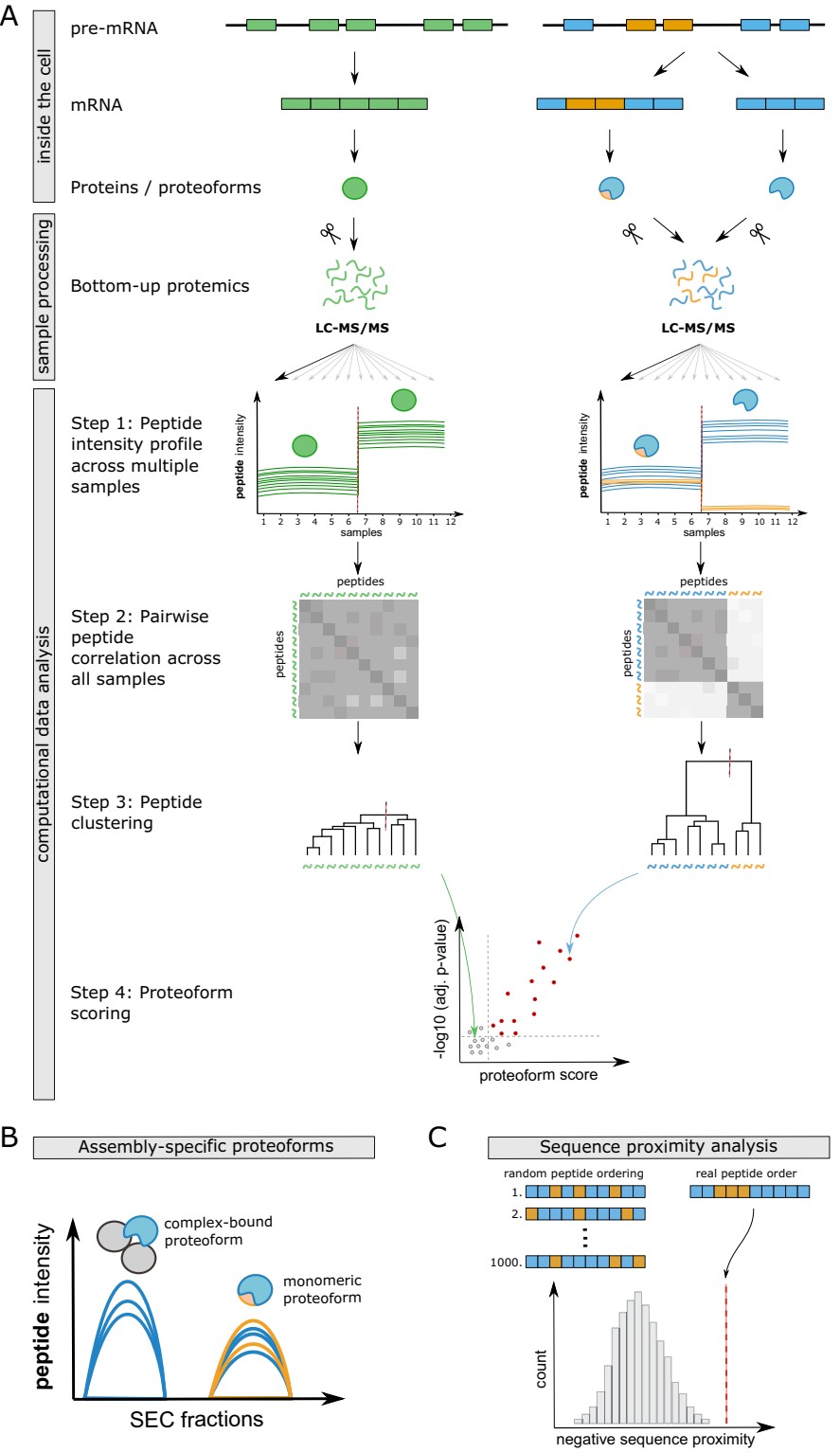

specifically designed for detecting proteoforms differing by a single peptide and could, therefore, achieve a convincing ROC curve. In the second benchmarking set, proteoform groups differed by two peptides, and COPF and PeCorA show similar ROC curves (Fig. 2 middle panel and Supplementary Fig. 1A). Here, COPF has slightly higher TPRs in the lower range of FPRs between 0 and 0.1. Finally, COPF markedly outperformed PeCorA in the third benchmarking set in which proteoform

groups differed by 50% of the protein's peptides (Fig. 2C right panel). The observed results are in line with expectations derived from the design of either tool. PeCorA is particularly sensitive in detecting single outlier peptides, whereas COPF is particularly powerful in detecting proteoforms that differ by multiple peptides. Looking at the assignment of peptides to proteoforms, it is apparent that PeCorA can successfully determine single proteoform peptides (orange color in Fig. 2C left panel).

**Fig. 1 Analysis concept and workflow overview. A** COPF is based on the concept that all peptides of a single proteoform protein that are differentially regulated across a dataset should follow the same quantitative pattern (left panel). In contrast, for differentially regulated proteins with more than one proteoform, peptides unique for each proteoform should follow a distinct quantitative pattern (right panel). In this figure, exemplary proteoforms generated via alternative splicing are depicted. Exons in the pre-mRNA are indicated as colored boxes and introns as lines. In the depicted example of alternative splicing, one splice variant only contains blue exons, thus resulting in a blue proteoform. The second splice variant additionally contains two orange exons, thus generating a mixed blue and orange proteoform. During proteolytic digestion for bottom-up proteomics, proteins and proteoforms are cleaved into peptides, a process during which the direct link between a peptide and its parental proteoform is lost. COPF takes advantage of datasets that assess quantitative peptide profiles across a large number of samples (Step 1). Here, this is illustrated by 12 samples measured across two different conditions. To quantify co-variance across samples, all pairwise peptide correlations are calculated for each protein (Step 2). The correlation distance (1 − Pearson's correlation) is subsequently used for hierarchical peptide clustering into two groups (Step 3). Finally, proteoform scores and corresponding *p*-values are calculated based on the within- vs. across-cluster Pearson's correlation (Step 4). Proteins without alternative proteoforms between conditions get a low score/high adjusted *p*-value (gray points), whereas proteins with multiple detected proteoforms between conditions achieve a higher proteoform score/ low adjusted *p*-value (red points). **B** COPF is embedded in the CCprofiler framework. Proteoform groups detected from SEC-SWATH-MS data can therefore directly be integrated into a protein complex analysis, to determine assembly-specific proteoforms. Here, peptides of the orange proteoform group are exclusively present in the monomeric state, whereas peptides of the blue proteoform group form a protein complex with the two proteins indicated in gray. **C** COPF has a post-processing module for sequence proximity analysis. Peptides are indicated as small boxes, colored by their assigned proteoform group in orange or blue.

---

However, mistakes increase with higher numbers of peptides per proteoform (violet color in Fig. 2C middle and right panel). In contrast, COPF can successfully group peptides to the correct proteoform group, given the proteoforms differ by minimally two peptides (orange color in Fig. 2C middle and right panel).

To assess the false discovery rate (FDR) estimation in COPF, we compared the multiple-testing adj. *p*-value with the empirical FDR in each of the ground truth benchmark datasets (Fig. 2D). The results show that FDR estimates by COPF are well-calibrated and overall conservative (generally following, but staying below the diagonal line) for proteoforms differing by two or more peptides (Fig. 2D middle and right panel). Introducing a proteoform score threshold in addition to the adj. *p*-values provides even more conservative results. Our analyses further show that the adj. *p*-values reported by PeCorA cannot be directly interpreted as FDRs. Although they represent an ordering relation and thus produce reasonable ROC curves (Fig. 2C), the adj. *p*-values of PeCorA do not correspond to protein-level FDR estimates (Fig. 2D). Although performing an additional multiple-testing correction of the *p*-values reported by PeCorA across all peptides in the dataset slightly reduced this effect (Supplementary Fig. 1A, B), additional measures would be necessary to derive FDR statistics for PeCorA.

In summary, the in silico benchmark analyses demonstrate that COPF can (1) confidently identify proteins with proteoforms, if proteoforms differ by two or more peptides, (2) correctly group peptides into the correct proteoform groups, and (3) provide well-calibrated and conservative FDR estimates for proteoform detection. The comparison with the state-of-the-art PeCorA tool shows that the two tools are complementary. Although PeCorA performs well for proteoforms differing by a single peptide, COPF outperforms PeCorA for proteoforms differing by multiple peptides.

**Identification of cell cycle- and assembly-specific proteoforms in a SEC-SWATH-MS dataset.** We applied the COPF strategy to our previously published native complex co-fractionation dataset of Hela CCL2 cells synchronized in interphase and mitosis[28] to identify cell cycle- and assembly-specific proteoforms. In this case, the dataset consists of native complexes isolated from cells in two cell cycle states, separated by SEC, and then analyzed by bottom-up proteomic analysis using DIA/SWATH-MS[21,31]. The workflow with individual steps is schematically illustrated in Fig. 3A. First, cells are lysed under close to native conditions to keep protein complexes intact. Second, the protein complex mixture is separated by SEC into 65 fractions. Third, each

sampled fraction is separately processed for bottom-up proteomic measurements by SWATH-MS[17] followed by peptide-centric analysis[32–34]. In a fourth step, peptide elution profiles across the SEC fractions are evaluated to infer protein complex assemblies by CCprofiler as described before[21,31] and additionally by the COPF method to identify proteoforms differentially associating with complexes within or across cell cycle states.

For the COPF analysis, we took the original peptide-level elution profiles from Heusel et al.[28] as a starting point. We imported the data into the CCprofiler framework for subsequent data preprocessing as follows. First, we annotated peptides with their start and end position in the canonical protein sequence. Second, peptides with overlapping start and end positions (e.g., peptides resulting from missed cleavages) were reduced to one representative peptide, selected based on the highest overall intensity across the dataset. Third, missing values were imputed for fractions with a valid value in both the preceding and following SEC fraction. Fourth, peptides detected in fewer than three consecutive fractions and peptides with zero variance along the SEC dimension were removed. Finally, only proteins with two or more remaining peptides were considered for further analysis by COPF. Importantly, intensities of individual peptides across all measured fractions and conditions were considered for calculating the pairwise correlations in COPF and to derive the scoring metrics.

The pseudo-volcano plot for the 5451 proteins of the SEC-SWATH-MS dataset that remained after the above filtering steps is shown in Fig. 3B. At a multiple-testing corrected *p*-value threshold of 10% and a minimal proteoform score of 0.1, COPF reported 317 proteins with functional proteoform groups. Two hundred and forty-three (77%) of these proteins are annotated with multiple isoforms in UniProt (Fisher's exact test: odds ratio = 1.8, *p*-value = $6 \times 10^{-7}$ when compared to the entire SEC-SWATH-MS dataset). A comparison with an independent study on phospho-signaling across cell cycle conditions[35] further showed that the 317 proteins are significantly enriched for cell cycle-regulated phosphosites (Fisher's exact test: odds ratio = 4.8, *p*-value = $7 \times 10^{-39}$). Proximity analysis of identified peptides within the protein sequence revealed that the proteoforms for 68 proteins (21%) were significantly closer in sequence proximity than expected by chance (*p*-value ≤ 10%) and proteoforms for an additional 92 proteins (29%) scored among the lowest 10% of possible *p*-values, given the number of peptides in the protein (for details, also see "Methods" section). We further analyzed the dataset with respect to proteoforms that associate with different complexes (assembly-specific) and to proteoforms that differ between cell cycle states (cell cycle specific).

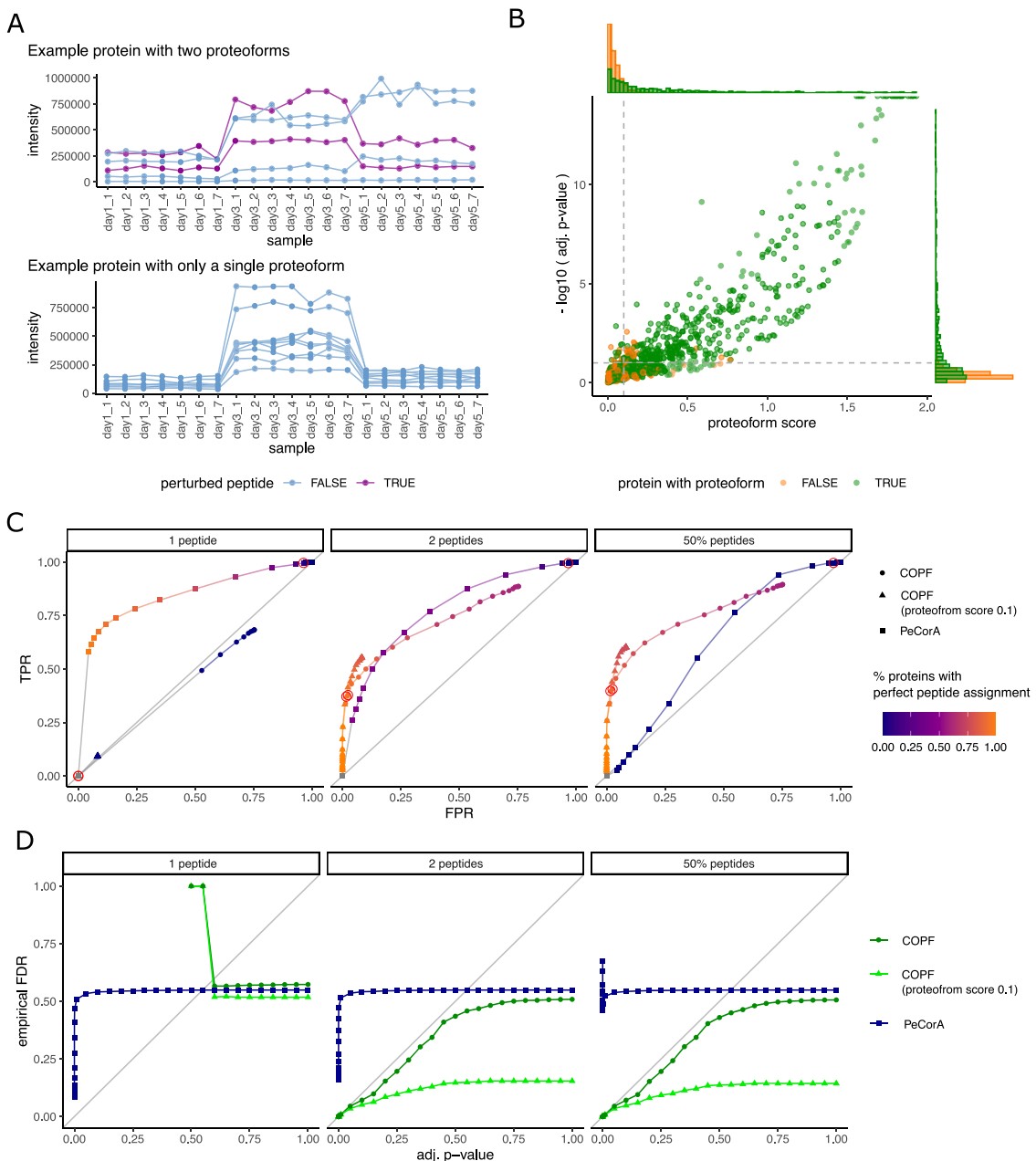

**Fig. 2 COPF benchmark. A** We generated an in silico benchmark dataset to evaluate COPF performance. Twenty-one replicate measurements of HEK293 cell lysates derived from the SWATH-MS interlab study[19] were selected and adjusted (i) to introduce variance across samples and (ii) to introduce proteoforms for a subset of proteins. The top panel shows the peptide-level intensity profiles for an exemplary protein consisting of two detectable proteoforms (two perturbed peptides in purple). The lower panel shows the peptide-level intensity profiles for a protein consisting of a single detectable proteoform (no perturbed peptides). **B** Pseudo-volcano plot for proteins with two (green) and only one (orange) proteoform. For this figure, proteoforms were generated by randomly perturbing between two peptides and 50% of the peptides in a protein. **C** Receiver operator characteristic (ROC) curves for three in silico benchmark datasets. They show the true positive rate (TPR) over the false positive rate (FPR). Individual points in the curve are generated by iterating over different adjusted *p*-value thresholds. The datapoints derived from an adjusted *p*-value threshold of 0.1 are highlighted by a red circle. **D** Empirical FDR values vs. estimated adjusted *p*-values. Individual points in the curves are generated by iterating over different adjusted *p*-value thresholds.

Proteoform groups for 109 proteins (34%) could be classified as assembly-specific, as we observed them in multiple distinct assembly states resolved along the SEC dimension (for details, see "Methods" section). In addition, COPF predicted proteoform groups of 124 proteins (39%) that were significantly differentially expressed between the two cell cycle stages (log2-fold change ≥ 1 and Benjamini–Hochberg (BH) adj. *p*-value ≤ 0.05). A summary of the proteoform characterization is provided in Fig. 3C.

As a final assessment of the global characteristics of the set of proteins detected as having multiple proteoform groups, we performed an enrichment analysis (Fig. 3D). We observed that the protein set is highly enriched in phosphoproteins and genes processed by alternative splicing. In addition, the protein set is enriched in UniProt keywords related to processes important during cell cycle progression, which is expected given the tested biological conditions.

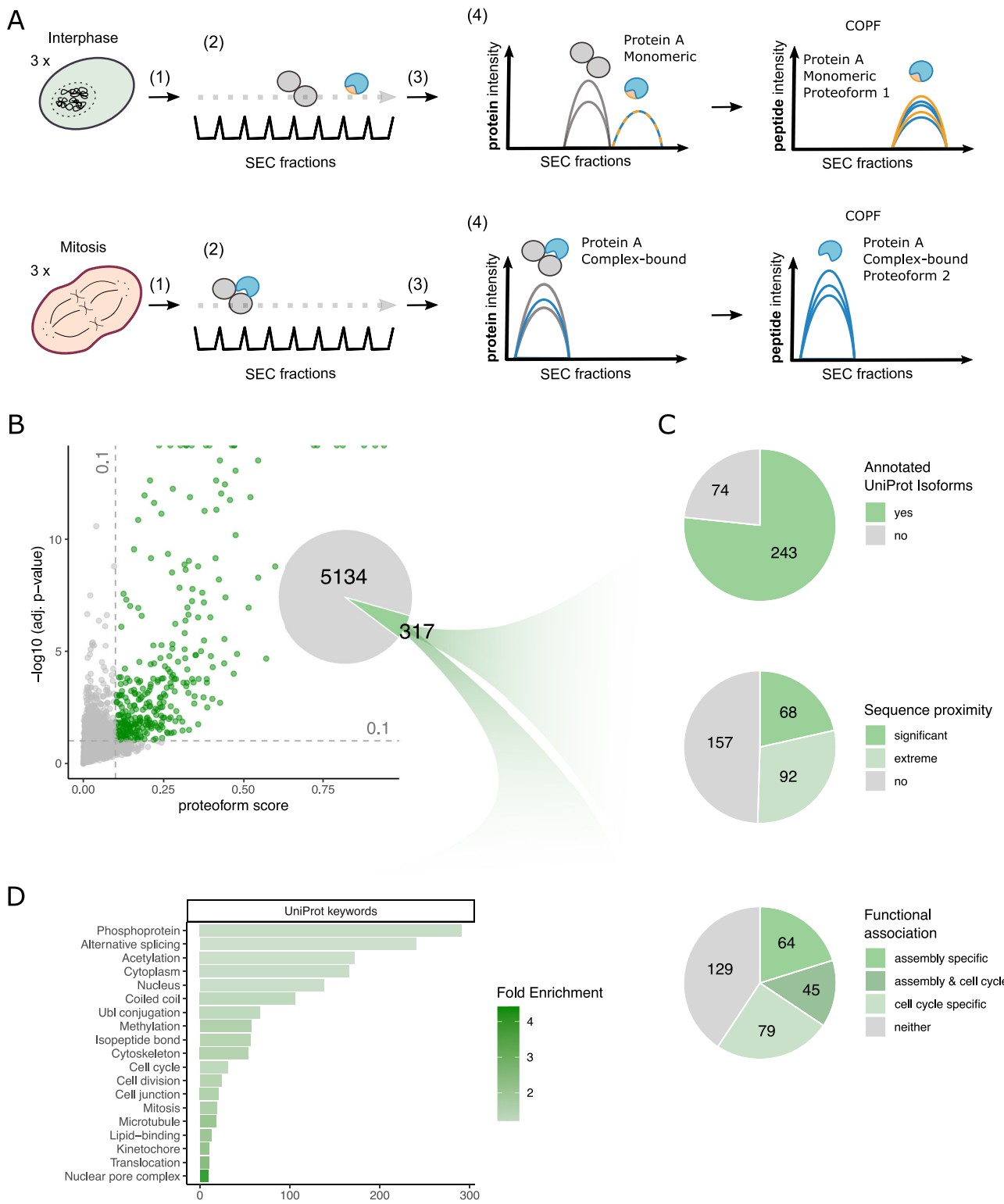

**Fig. 3 Global insights into cell cycle- and assembly-specific proteoforms in SEC-SWATH-MS data. A** Schematic overview of the experimental design and SEC-SWATH-MS analysis workflow. HeLa cells were blocked in either interphase or mitosis. Samples were subject to mild cell lysis (1) followed by protein complex fractionation via SEC (2). Each of the consecutively sampled fractions was separately subjected to a bottom-up proteomic workflow including tryptic digestion and MS data acquisition by SWATH-MS (3). The resulting protein- and peptide-level quantitative profiles along the SEC dimension were used for protein complex assessment and proteoform analysis with COPF (4). **B** Pseudo-volcano plot showing the proteoform scoring results for the cell cycle SEC-SWATH-MS dataset. At a multiple-testing corrected *p*-value threshold of 10% and minimal proteoform score of 0.1, a set of 317 proteins was predicted by the algorithm to contain multiple proteoform groups. **C** Assessment of the reported proteoform groups. **D** UniProt keyword enrichment analysis of the predicted set of reported proteoform groups.

In addition to these global insights, our dataset provided a rich source of new biological information. In the following, we present selected examples that highlight different mechanisms generating proteoforms and different functional associations of proteoforms with either the cell cycle or protein assembly.

The first example is the proteasome subunit-β type-7 (PSMB7, UniProt ID: Q99436), which is convincingly resolved in our SEC data. The proteasome assembly line is a well-studied yet still heavily investigated system. Figure 4A shows a simplified schematic of the process. A key step in the prevalent model of 20S particle assembly is the integration of PSMB7 as the last β-subunit, triggering proteolytic cleavage of its pro-peptide, followed by formation of the full 20S core proteasome complex[36]. Our data-driven COPF strategy identified two assembly-specific proteoforms for PSMB7 (also see Supplementary Fig. 2A). The SEC profiles of all detected PSMB7 peptides are shown in Fig. 4B. The peptides assigned to the two different proteoform groups are highlighted in blue and orange. Although the peptides of both proteoform groups participate in the lower molecular weight (MW) peak around fraction 33, only peptides of the orange proteoform group were detected in the higher MW peak around fraction 25. From protein co-elution analysis (Supplementary Fig. 2C) and our previous study[21], we know that the peak group around fraction 33 corresponds to a proteasome assembly intermediate and the peak group around fraction 25 corresponds to the full 20S core proteasome. Checking the location of the detected peptides along the PSMB7 sequence reveals that the peptides of the blue proteoform correspond to the two N-terminal peptides (Fig. 4C, also see Supplementary Fig. 2B). The second peptide (TGTTIAGVVYK) spans the known proteolytic cleavage site of the PSMB7 pro-peptide. To verify our finding, we performed a targeted re-extraction of the semi-tryptic peptide (TTIAGVVYK) that is produced by tryptic digestion of the processed, short proteoform using Skyline[37,38]. The extracted signal of this semi-tryptic peptide is highlighted in green in Fig. 4B, C (also see Supplementary Fig. 3). In contrast to the fully tryptic (blue) peptide, the cleaved peptide sequence co-elutes with the peptides of the orange proteoform group, indicating that the processed form is integrated in the 20S proteasome core complex as expected and identifying the precise location of the proteolytic processing that generates the proteoform.

The nuclear pore complex (NPC) protein Nup98-Nup96 (UniProt ID: P52948) presents a second example of the capacity of COPF for data-driven proteoform assignment. The Nup98-Nup96 proteoform groups were identified by the algorithm as both assembly and cell cycle specific (Fig. 4D). The *Nup98* gene is known to encode a 186 kDa precursor protein that undergoes autoproteolytic cleavage, to generate a 98 kDa nucleoporin (NUP98) and a 96 kDa nucleoporin (NUP96) (Fig. 4D)[39–42]. NUP96 is an important scaffold component of the NPC, whereas NUP98 has diverse functional roles during mitosis. Previously, we showed the upregulation of the Nup107-160 subcomplex (Corum ID: 87) in mitosis as compared to interphase (see Fig. 6H in ref. [28]). There, we stated that the protein product of the *Nup98* gene is upregulated together with the other members of the Nup107-160 subcomplex, i.e., proteins SEC13 (P55735), NUP107 (P57740), NUP160 (Q12769), NUP43 (Q8NFH3), NUP37 (Q8NFH4), NUP133 (Q8WUM0), SEH1L (Q96EE3), and NUP85 (Q9BW27). Here, our purely data-driven approach was capable of correctly grouping peptides according to the two known proteoforms NUP98 and NUP96 (Fig. 4E, F, also see Supplementary Fig. 2D, E). Based on this assignment, we can now also demonstrate that only the NUP96 proteoform integrates into the Nup107-160 subcomplex and follows its behavior across the cell cycle (Supplementary Fig. 2F). This is in line with reports from previous studies[43].

In addition to the above-mentioned examples where proteoforms were derived from enzymatic or autoproteolytic cleavage, the COPF algorithm could further detect proteoforms derived from alternative splicing, exemplified by the nuclear autoantigenic sperm protein (NASP, UniProt ID: P49321). Previous studies reported two alternative-splicing-derived proteoforms of the NASP protein that can be detected in transformed cell lines: a somatic form (sNASP) and a shorter, testicular form (tNASP)[44]. In our HeLa cell cycle dataset, COPF detected two assembly-specific proteoform groups that correctly matched the annotated proteoforms sNASP and tNASP (Fig. 5A, also see Supplementary Fig. 4A). The two distinct peptide peak groups in different MW regions of the SEC separation indeed confirm previous findings that the two proteoforms engage in different assemblies.

Whereas the examples described above refer to well-annotated proteoforms that we were able to resolve without including prior knowledge in the analysis, our findings also uncovered less well-understood proteoforms. Figure 5C, D show the profile and sequence location for peptides of Transmembrane protein 106B (TMEM106B, UniProt ID: Q9NUM4), which has no annotated sequence variants or specific post-processing steps annotated in UniProt. Nevertheless, COPF identified two clearly distinguishable, assembly-specific proteoforms (Supplementary Fig. 4B). In recent literature, we found evidence that TMEM106B is a lysosomal membrane protein that, upon membrane integration, undergoes evolutionarily conserved regulated intramembrane proteolysis[45,46]. This involves a two-step mechanism where the luminal domain is first cleaved off by an unknown enzyme at AA residue 127, followed by a second cleavage (at AA 106) of the N-terminal fragment that is still anchored in the membrane (cleavage sites are indicated by the scissors in Fig. 5D). Our data suggest that the peptides of the blue proteoform group belong to the C-terminal luminal domain, eluting separately in the low MW range, likely consistent with a monomeric form (expected monomeric MW = 31 kDa, observed elution at ~42 ± 10 kDa as estimated based on the log-linear MW calibration of the SEC fractions, see "Methods" section). The high MW signal likely corresponds to the full protein integrated into the lysosomal membrane. Enrichment analysis of co-eluting proteins in the high MW range shows an enrichment in membrane-associated proteins (Supplementary Fig. 4C, D).

The strong enrichment of phosphoproteins in general and more specifically of proteins with previously reported cell cycle-regulated phosphosites[35] in the set of proteins with proteoform groups was a significant finding in the global analysis of COPF results (Fig. 3D). To follow-up on these results, we set out to identify the specific proteoform groups that are enriched in cell cycle-regulated phosphosites. For this, we performed enrichment analyses in each of our predicted proteoform groups to test whether they are enriched for cell cycle-regulated phosphosites as determined by Karayel et al.[35]. In total, 42 out of the 317 proteins with multiple proteoform groups (13%) had one proteofrom group significantly enriched in previously annotated cell cycle-regulated phosphosites, thus suggesting that these proteoforms might be phosphorylation-derived. Twenty-eight (67%) of these proteins also turned out to be cell cycle stage-specific (log2-fold change ≥ 1 and BH adj. *p*-value ≤ 0.05, Fig. 5E). One such example is the Ataxin-2-like protein (UniProt ID: Q8WWM7). Peptides of the first proteoform group (orange) are higher-abundant in mitosis (log2-fold change = 1.38 and BH adj. *p*-value = 0.002), whereas the second proteoform (blue) is significantly lower-abundant (log2-fold change up to −2.66 and BH adj. *p*-value = 0.005) (Fig. 5F and Supplementary Fig. 5). Four out of five regulated phosphosites[35] (Fig. 5G) fell into sequence regions covered in our dataset, exactly matching the four peptides of the second proteoform (blue). Making use of the possibility to re-extract data from SWATH-MS maps, we performed a targeted

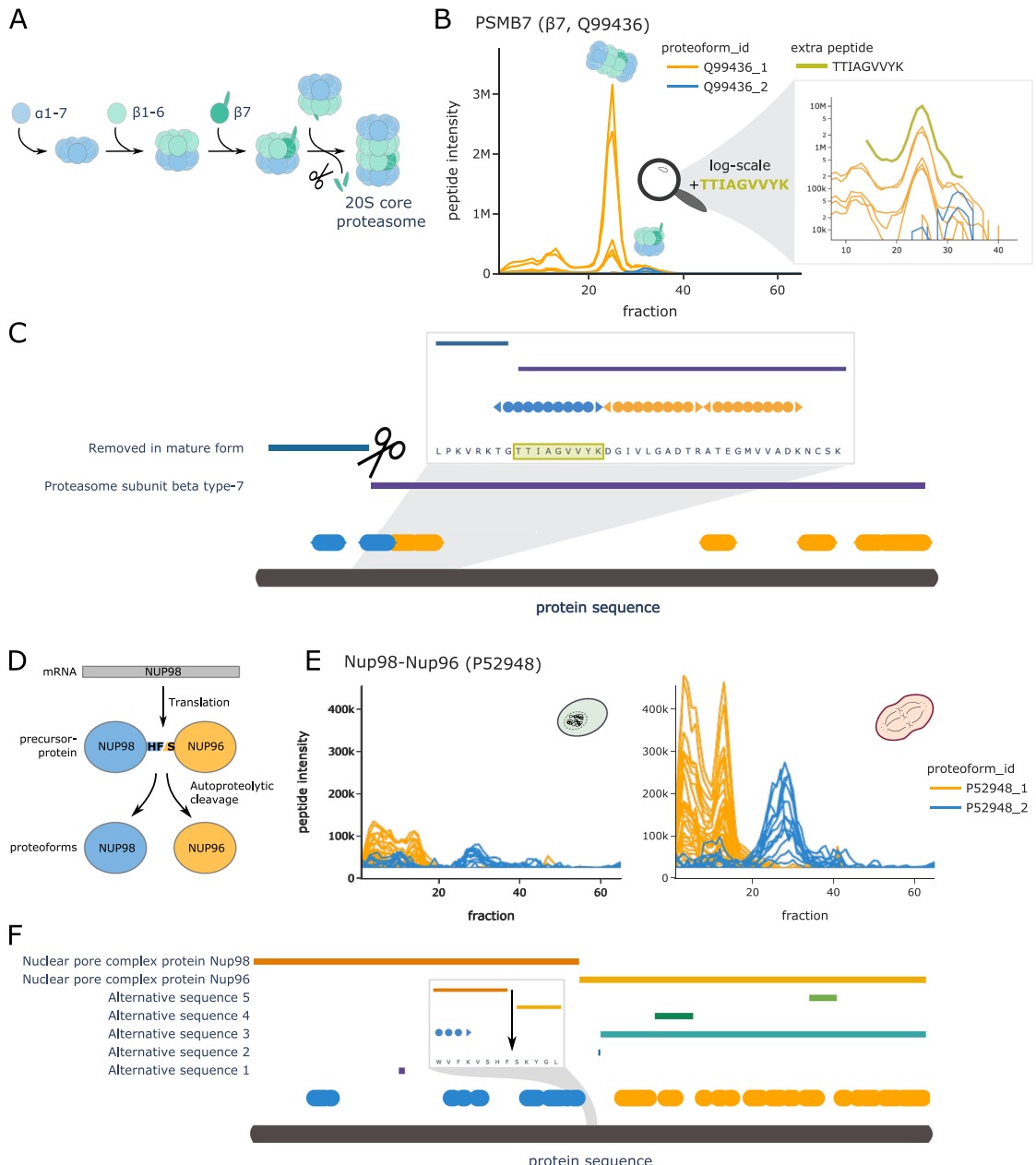

**Fig. 4 COPF results for PSMB7 and NUP98/96. A** Schematic overview of the proteasome assembly line. **B** Peptide profiles of the proteasome subunit-β type-7 (β7, PSMB7, UniProt ID: Q99436) in interphase. Peptides of the two assigned proteoform groups are colored in orange and blue. A zoom-in on fraction 10 to 40 is shown in log scale, including an additional semi-tryptic peptide TTIAGVVYK that represents the N-terminal tryptic peptide of the processed proteoform (green). Please note that the abundance values between the orange and blue peptides cannot be directly compared to the semi-tryptic peptide TTIAGVVYK in green because of the separate analysis platforms used. **C** Protein sequence plot for PSMB7. Sequence coverage and position of the detected peptides of the two assigned proteoform groups are indicated in blue and orange. Pro-peptide and chain information from UniProt are indicated as horizontal bars. The known pro-peptide cleavage site is indicated by scissors. A zoom-in representation is provided for the region around the annotated cleavage site. The semi-tryptic peptide from **B** is highlighted in green. **D** Schematic overview of the NUP98 and NUP96 proteoform biogenesis from the same mRNA and precursor protein (UniProt ID: P52948). **E** Peptide profiles of the *Nup98* gene product in interphase (left) and mitosis (right). Peptides of the two assigned proteoform groups are colored in orange and blue. **F** Protein sequence plot for the *Nup98* gene product. Sequence coverage and position of the detected peptides of the two assigned proteoform groups are indicated in blue and orange. Chain information and alternative sequence isoforms from UniProt are indicated as horizontal bars. The known autocatalytic cleavage site of NUP98/96 is shown as an arrow in a zoom-in on the relevant sequence region.

analysis in Skyline to detect and quantify the expected phosphopeptides in our dataset (Supplementary Fig. 6). As predicted, we could confirm mitosis-specific phosphorylation for two of the four peptides (EIESS[+80]PQYR and TLSS[+80]PSNRPSGETSVPPPPAVGR, Fig. 5F). One phosphopeptide (GPPQS[+80]PVFEGVYNNSR) had only a weak signal (purple) and the fourth could not be detected. Nevertheless, it is remarkable that the phosphopeptides could be detected and quantified in the samples given the long processing protocol without enrichment and without specific phosphatase inhibition treatment. These findings highlight that the COPF approach is capable of determining phospho-specific proteoform groups,

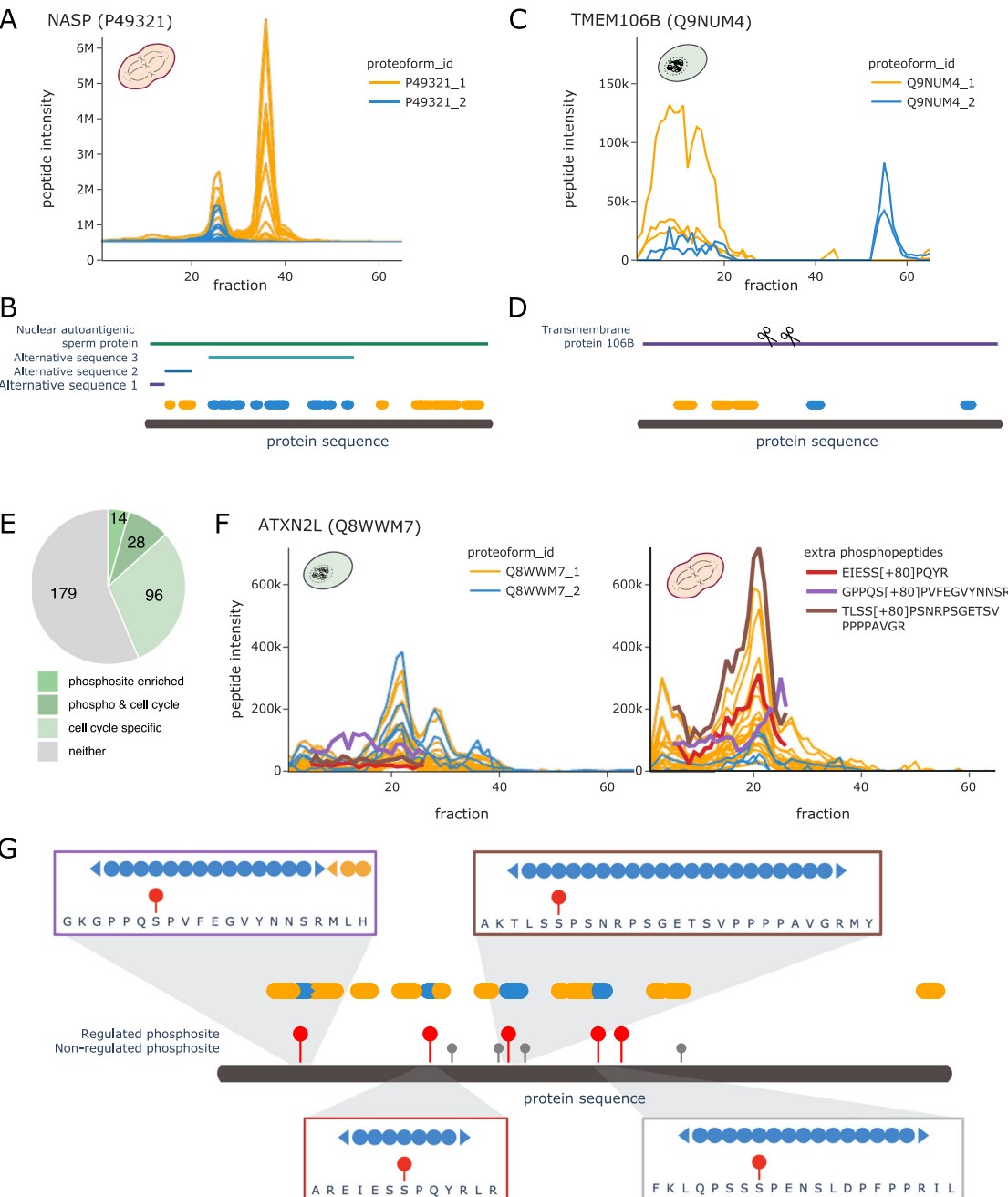

**Fig. 5 COPF results for NASP, TMEM106B, and ATXN2L. A** Peptide profiles of the nuclear autoantigenic sperm protein (NASP, UniProt ID: P49321) in mitosis. Peptides of the two assigned proteoform groups are colored in orange and blue. **B** Protein sequence plot for NASP. Sequence coverage and position of the detected peptides of the two assigned proteoform groups are indicated in blue and orange. Chain information and alternative sequence isoforms from UniProt are indicated as horizontal bars. **C** Peptide profiles of the transmembrane protein 106B (TMEM106B, UniProt ID: Q9NUM4) in interphase. Peptides of the two assigned proteoform groups are colored in orange and blue. **D** Protein sequence plot for TMEM106B. Sequence coverage and position of the detected peptides of the two assigned proteoform groups are indicated in blue and orange. Chain information from UniProt is indicated as horizontal bar. Suggested catalytic cleavage sites[45,46] are indicated by scissors. **E** Pie chart illustrating the number of proteins with a proteoform enriched in cell cycle-regulated phosphosites[35] and their overlap with cell cycle specificity of the proteoform. **F** Peptide profiles of the Ataxin-2-like protein (ATXN2L, UniProt ID: Q8WWM7) in interphase (left) and in mitosis (right). Peptides of the two assigned proteoform groups are colored in orange and blue. In addition, traces of subsequently extracted phosphopeptides are highlighted in red, violet, and brown. Please note that the abundance values between the orange and blue peptides cannot be directly compared to the phosphopeptides because of the separate analysis platforms used. **G** Protein sequence plot for ATXN2L. Sequence coverage and position of the detected peptides of the two assigned proteoform groups are indicated in blue and orange. Cell cycle-regulated phosophosites are indicated by red sticks, whereas non-regulated phosphosites are indicated in gray[35]. Regions of individual phosphopeptides are visualized in separate zoom-in windows, matching the colors of the respective peptides in **F**.

given that at least two peptides are involved. Here, it is important to emphasize again that the strategy by which phospho-specific proteoform groups were detected with COPF directly links these detections to their biological relevance in the cell cycle.

**Tissue-specific proteoforms in SWATH-MS data of different mouse tissues**. In comparison to the SEC-SWATH-MS dataset where native protein complexes are separated and analyzed in consecutive fractions, bottom-up proteomic datasets of unfractionated samples are more commonly available. Using a previously published SWATH-MS dataset of five different mouse tissues from eight BXD mice each[29] (Fig. 6A), we tested the assumption that the COPF strategy to identify functional proteoforms was also applicable to peptide intensity vs. sample data matrices from sample sets of sufficient size and variability between samples.

We initially imported the peptide-level data matrix (intensity vs. sample) into the CCprofiler framework and applied the same data processing steps as for the application of COPF for SEC-SWATH-MS data described above, except for the consecutive identification filter and missing value imputation. Proteins with two or more remaining peptides were considered for further analysis by COPF and resulted in the pseudo-volcano plot for 2885 proteins shown in Fig. 6B. At a multiple-testing corrected $p$-value threshold of 10% and a minimal proteoform score of 0.1, COPF reported 63 proteins with potential functional proteoform groups. Fourteen (22%) of these are annotated with multiple isoforms in UniProt. Peptide proximity analysis within the coding sequence further revealed that the proteoforms for 19 proteins (30%) were significantly closer in sequence proximity than expected by chance ($p$-value ≤ 10%) and proteoforms for an additional 7 proteins (11%) scored among the lowest 10% of possible $p$-values, given the number of peptides identified for the respective protein (for details, also see "Methods" section). Further evaluation of the detected proteoform groups revealed that proteoform groups for 56 proteins (89%) could be classified as tissue specific. This classification is based on the protein being differentially regulated when using tissue and predicted proteoform group information as prior knowledge for an analysis of variance (ANOVA) (Bonferroni corrected $p$-value ≤ 0.01). A summary of the proteoform characterization is provided in Fig. 6C.

We further performed an enrichment analysis of the proteins annotated with multiple proteoforms (Fig. 6D), showing that they are significantly enriched in keywords related to fatty acid metabolism. Interestingly, many proteins are also associated with acetylation.

Finally, we compared the proteoform containing proteins called by COPF with those determined by PeCorA[27] (Fig. 6E). In total, PeCorA reported significant peptides for 2730 out of 2885 proteins (95%) at an adj. $p$-value cutoff of 10%. This set covers all but one proteoform containing protein determined by COPF. If an additional multiple-testing correction of the adj. $p$-values reported by PeCorA was performed across all peptides, the number of proteins reported by PeCorA dropped only by 41, still covering 93% of all proteins. These findings are in line with the benchmarking results (see above), suggesting that PeCorA is more sensitive than COPF at determining outlier peptides, however at the cost of a potentially high FPR.

Among the 63 proteins assigned to have functional proteoform groups by COPF, multiple biologically interesting instances stood out, exemplified by LIM domain-binding protein 3 (Ldb3, also known as Cypher, UniProt ID: Q9JKS4), and sorbin and SH3 domain-containing protein 2 (Sorbs2, UniProt ID: Q3UTJ2), respectively.

The protein Ldb3 was previously described as muscle specific[47] and, accordingly, was found to be highly expressed only in the heart and quadriceps samples of our dataset (Fig. 7A). The COPF strategy clearly assigned the peptides of Ldb3 into two tissue-specific proteoform groups, indicated in orange and blue (Fig. 7B). These proteoforms directly match the previously annotated splice variants of Ldb3, where peptides of the first proteofrom group, Q9JKS4-1, exactly map to the canonical sequence region that also has an alternative sequence variant (alternative sequence 1 in Fig. 7C). To further validate this finding, we performed a targeted extraction of peptides in the alternative sequence region (peptides: VVANSPANADYQER and FNPSVLK) and could confirm their expression in quadriceps tissue (Supplementary Fig. 7). These findings are in line with previous studies that reported tissue-specific expression of the alternative splice variant in the skeletal muscle[47].

For Sorbs2 (Fig. 7D), our fully data-driven approach assigned the peptides to two clearly distinct proteoform groups (Fig. 7E). The peptides of the orange proteoform group are abundant in the brain, heart, and liver tissues, whereas the peptides of the second, blue proteoform group are observed exclusively in the brain. Mapping the identified peptides to the canonical protein sequence and matching them with annotated sequence variants (Fig. 7F), it can be observed that the two blue peptides map to the region of a brain-specific splice variant (alternative sequence 10) that includes an exon that has previously been shown to only be expressed in the brain tissue and exclusively in the neurons[48–50].

## Discussion

Until recently, proteoforms have mainly been inferred from transcriptomics data of next-generation sequencing studies that aim at discovering different mRNA transcripts of the same gene, mostly generated by alternative splicing. In these studies, it is assumed that the alternative transcripts are further translated to protein sequences, which is not always the case[51,52]. One approach to study proteoforms directly on the protein level is to generate and apply proteoform-specific antibodies, which can only be achieved at fairly low throughput and high cost. One recent example represents a study that characterized alternative CD6 isoforms generated by alternative splicing[53]. However, the systematic identification of proteoforms on a proteome-wide scale has only recently been enabled by MS-based technologies.

Impressive progress in top-down proteomics has led to the identification and characterization of a few thousand proteoforms in parallel[13,14]. However, the proteomic coverage of top-down proteomics approaches is still technically limited[12,15]. As the connection between protein and peptide is lost at an early stage of bottom-up proteomic workflows, this method has mostly been considered unsuitable for the analysis of proteoforms to date. In this study, we present COPF, an analysis concept and software for proteoform detection in bottom-up proteomics data. COPF has multiple unique characteristics that distinguish it from other methods for proteoform detection in bottom-up proteomics data: first, COPF is applicable to complex experimental designs that do not follow classical two- or multi-condition comparisons and it is also applicable to data generated from a single sample or biological condition exemplified by a single SEC-SWATH-MS experiment. Second, COPF is optimized for proteoforms differing by multiple peptides, e.g., derived from alternative splicing, proteolytic cleavage, truncation or multiple co-regulated PTMs. Third, COPF includes a statistical model to estimate an FDR for proteoform detection. Fourth, post-analysis modules such as peptide proximity analysis evaluate the biological credibility of the determined proteoforms. Finally, COPF is embedded in the CCprofiler library that we originally designed for the analysis of

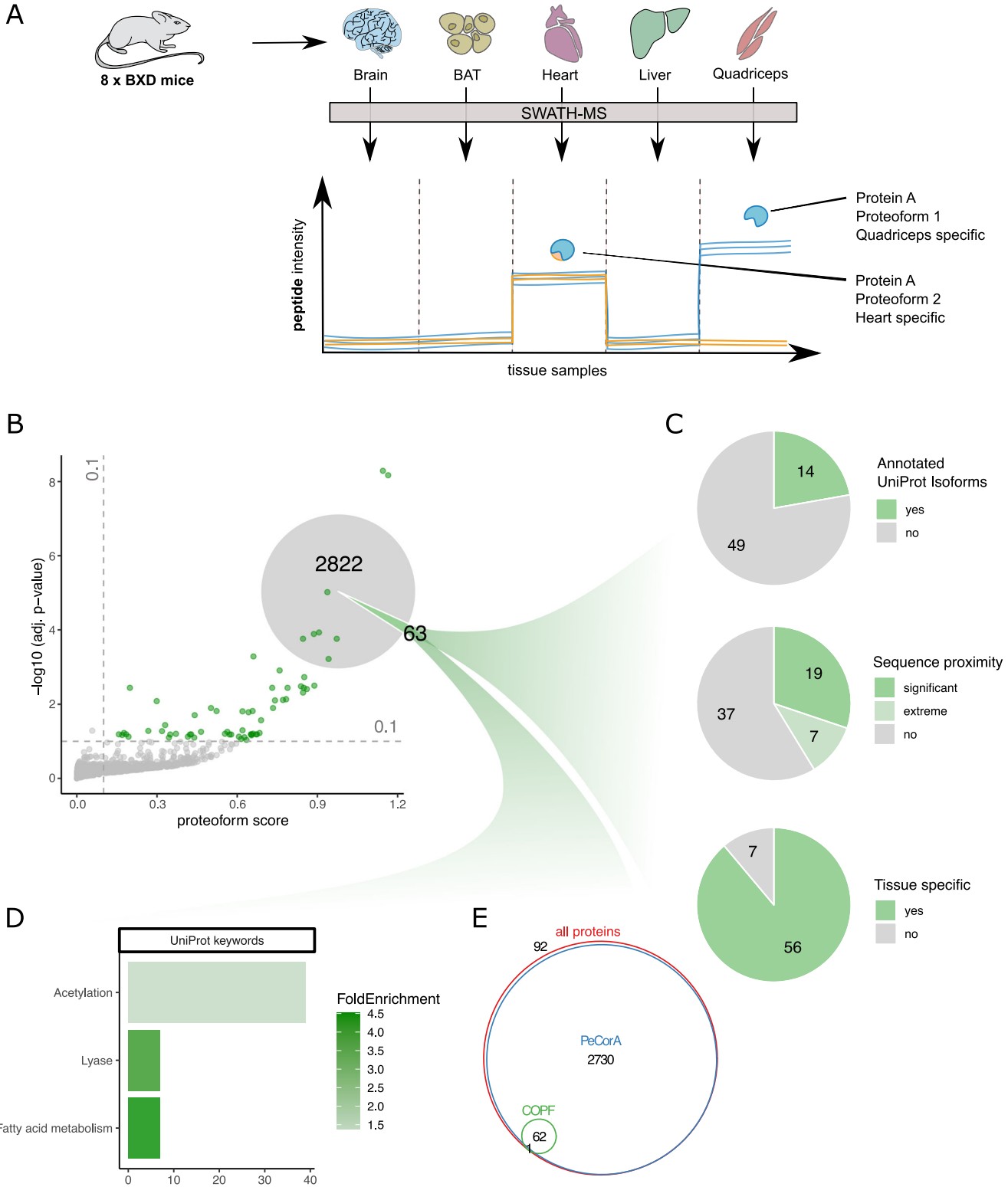

**Fig. 6 Global insights into tissue-specific proteoforms in full-proteome SWATH-MS data. A** Schematic overview of the experimental design and analysis concept. Tissue samples were obtained from eight different strains of the BXD mouse genetic reference panel. The selected tissues were the brain, brown adipose tissue (BAT), heart, liver, and quadriceps. Each sample was separately processed by bottom-up proteomics using SWATH-MS. The resulting peptide-level quantitative profiles can be used to identify tissue-specific proteoform expression, here exemplified by muscle-specific expression of a protein with a heart-specific proteoform (blue and orange) and a quadriceps-specific proteoform (only blue). **B** Pseudo-volcano plot showing the proteoform scoring results for the mouse tissue SWATH-MS dataset generated by the COPF algorithm. At a multiple-testing corrected *p*-value threshold of 10% and minimal proteoform score of 0.1, 63 proteins were predicted with multiple proteoform groups. **C** Assessment of the reported proteoform groups. **D** UniProt keyword enrichment analysis of the 63 proteins with multiple reported proteoform groups. **E** Venn diagram showing the overlap of proteoform containing proteins determined by COPF (10% adj. *p*-value and proteoform score ≥ 0.1) and PeCorA (10% adj. *p*-value) compared to the total set of proteins in the dataset.

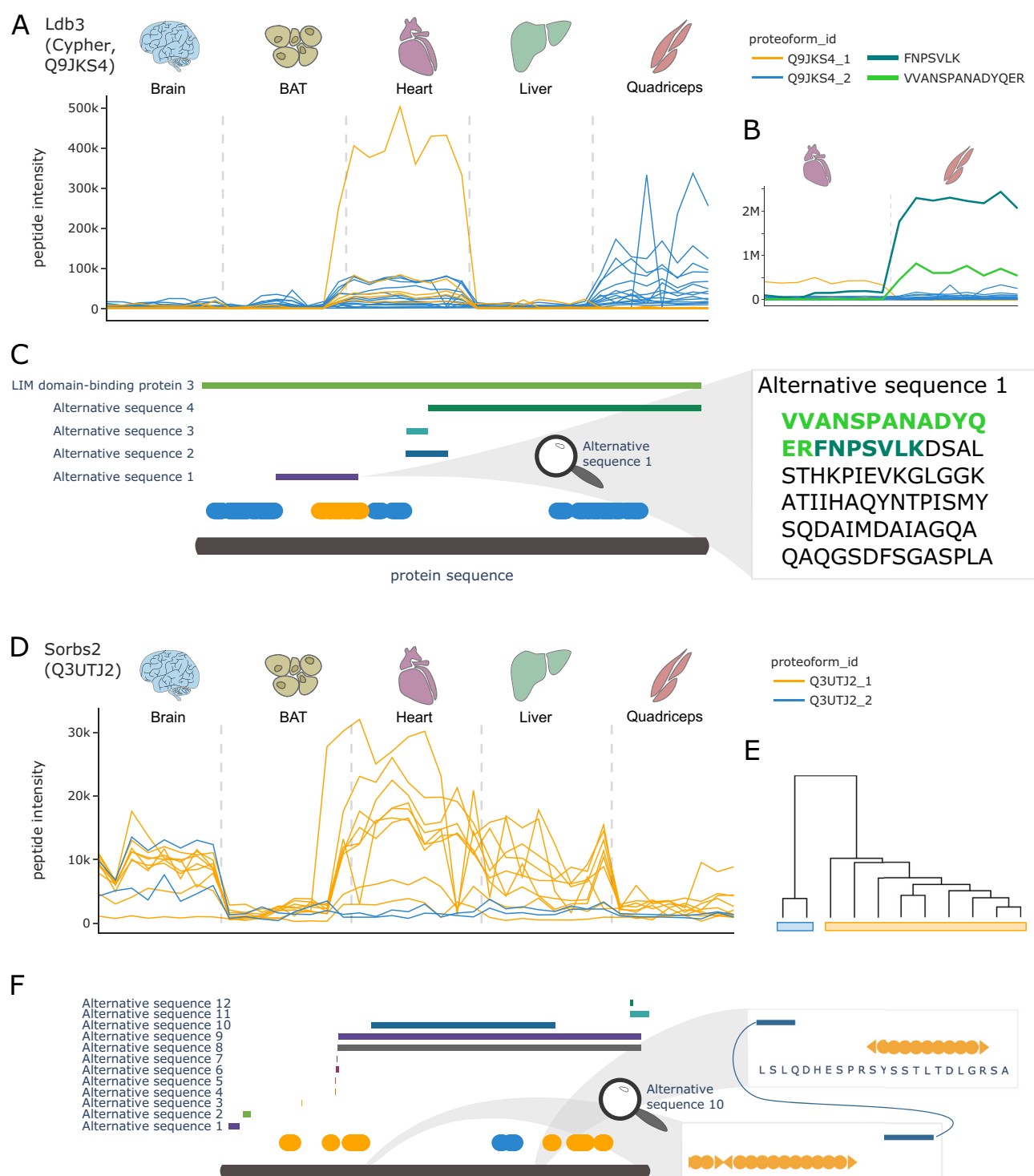

**Fig. 7 COPF results for Ldb3 and Sorbs2. A** Peptide profiles of the LIM domain-binding protein 3 (Ldb3, also known as Cypher, UniProt ID: Q9JKS4). Peptides of the two assigned proteoform groups are colored in orange and blue. **B** Zoom-in on muscle tissue, specifically highlighting two additionally extracted peptides VVANSPANADYQER (light green) and FNPSCLK (dark green) that are specific to the known skeletal muscle-specific splice isoform of Ldb3. Please note that the abundance values between the orange and blue peptides cannot be directly compared to the additionally extracted peptides in green because of the separate analysis platforms used. **C** Protein sequence plot for Ldb3. Sequence coverage and position of the detected peptides of the two assigned proteoform groups are indicated in blue and orange. Chain information and alternative sequence isoforms from UniProt are indicated as horizontal bars. The zoom-in on alternative sequence 1 shows the sequence of the skeletal muscle-specific splice isoform. The two peptides from **B** are highlighted in light and dark green, respectively. **D** Peptide profiles of the sorbin and SH3 domain-containing protein 2 (Sorbs2, UniProt ID: Q3UTJ2). Peptides of the two assigned proteoform groups are colored in orange and blue. **E** Clustering dendogram for Sorbs2. **F** Protein sequence plot for Sorbs2. Sequence coverage and position of the detected peptides of the two assigned proteoform groups are indicated in blue and orange. Alternative sequence isoforms from UniProt are indicated as horizontal bars. A zoom-in on the terminal regions of alternative sequence 10 shows that the orange proteoform does not cover this sequence region.

protein complexes in SEC-SWATH-MS datasets. COPF results can therefore be directly integrated into protein complex analysis to determine assembly-specific proteoforms, a question that could not be systematically assessed before.

A key challenge during the development of COPF was the lack of a ground truth dataset with known functional proteoforms. To nevertheless carry out a performance evaluation of our software, we generated an in silico reference dataset. Our analyses demonstrate that COPF can identify proteins with multiple proteoforms and assign detected peptides to the correct proteoform at well-controlled error rates (Fig. 2). We further benchmarked COPF performance against PeCorA, a recently published, state-of-the-art tool, for the detection of outlier peptides and potential proteoforms from bottom-up proteomics data[27]. Our analysis showed that COPF outperforms PeCorA in cases where proteoforms differ by two or more peptides. Based on the conceptual design of COPF, it cannot detect proteoforms differing by a single peptide. PeCorA can identify these single peptides with high sensitivity, however coming at the cost of lower selectivity, which is currently not controlled on the protein level. This effect could be observed in both the benchmarking and mouse tissue dataset. Overall, it is difficult to distinguish technical outliers from true biological signals among the PeCorA results. Proteoforms reported by COPF require at least two peptides differing between proteoforms, thus reducing the chance of observing a purely technical artifact. In addition, the grouping of peptides by COPF enables the direct report of proteoform groups. PeCorA only reports single peptides differing in intensity between samples but does not derive conclusions on whether multiple outlier peptides of the same protein co-vary because they are derived from the same proteoform or from different proteoforms. The difference in COPF and PeCorA performance, depending on the number of peptides that are required to differentiate between proteoforms, suggests that PeCorA is particularly suitable to detect proteoforms derived by single PTMs. In contrast, COPF is particularly sensitive in detecting proteoforms covering larger sequence stretches, such as those generated by proteolytic cleavage events or splice isoforms.

The performance characteristics observed in the benchmarking analyses suggest that COPF sensitivity will increase markedly with technical advances that improve sequence coverage as well as data completeness across samples. In this study, COPF was applied to two datasets generated by SWATH-MS. When compared to more classical data-dependent acquisition, DIA (or SWATH-MS) has the advantage of both increased data completeness and quantitative accuracy[54]. Recent developments on MS instrument level as well as in data acquisition and analysis promise further improvements with regard to proteome and sequence coverage[55,56].

In addition to considerations on proteoform types detectable by a given analysis approach, experimental design is an important factor that determines which strategy is suitable for a given study. PeCorA is designed for multi-condition comparisons with well-defined and fairly homogeneous groups, whereas COPF is specifically tailored towards complex study designs. COPF is applicable to continuous datasets such as those generated by SEC-SWATH-MS and also to nested study designs with multiple covariates. Due to these conceptual differences, PeCorA could not be applied to the SEC-SWATH-MS dataset presented herein. We expect that the primary applications of COPF will be datasets including highly heterogeneous samples for which a large degree of overall quantitative variation is observed. Both the differential SEC-SWATH-MS dataset and the mouse tissue dataset presented herein provide good examples for data with high abundance variation. Other promising study designs for COPF analysis could, e.g., be linked to sub-cellular localization maps or similar

systems with a high degree of biological diversity. Applications might also extend to less well-defined study designs by mining large proteomic datasets available from proteomic databases such as Pride[57]. However, the scope of possible applications and its limitations still remain to be explored.

Results of the COPF analysis presented herein are based on splitting the peptides of a protein into maximally two proteoform groups. However, it is expected that some proteins might have more than two functionally relevant proteoforms that could, with this approach, not be resolved. To address this limitation, alternative clustering strategies that can group peptides into multiple proteoform groups are also available in COPF and can be explored by users who aim to gain a more fine-grained separation of proteins into a variable number of proteoform groups (also see "Methods" section). Such analyses will strongly benefit from a higher sequence coverage, which is expected to be achieved by newest DIA technologies in the near future[55,56].

An important distinction between functional proteoform groups assigned by COPF and those determined by top-down proteomics approaches is that COPF does not fully characterize the proteoform's complete primary AA sequence and all of its modifications. It merely determines whether peptides exist that can differentiate the different biological contexts of a protein. Importantly, proteoform groups detected by COPF can directly imply a functional consequence depending on the study design. The power of the method is thereby directly linked to specific dataset properties. SEC-SWATH-MS data, as presented in this study, e.g., provides the opportunity to link the detection of proteoform groups directly to protein complex assembly, a property that makes such datasets unique and especially interesting for systematic proteoform investigation by COPF. A second factor based on which COPF can detect proteoform groups and distinguish functional associations is the biological context of a study, namely from the different biological conditions at hand. In the two datasets presented herein, these correspond to cell cycle- and tissue-specific proteoform groups. The focus of COPF to detect proteoforms with different functionality therefore addresses the increasing discrepancy between the ability of high-throughput data acquisition techniques to identify new chemical entities and the challenge to associate functional significance to these newly discovered entities.

The presented examples in this study include proteoform groups generated by proteolytic cleavage (Figs. 4A–C and 5C, D), autocatalytic cleavage (Fig. 4D–F), alternative splicing (Figs. 5A, B and 7A–F), and multiple phosphorylations (Fig. 5F, G). These examples demonstrate that the proposed strategy is, in principle, agnostic to the different mechanisms by which proteoforms can be generated inside the cell. This is in stark contrast to most other approaches that commonly investigate a subset of mechanisms. On one hand, proteoforms originating from alternative splicing are most commonly studied by proteogenomic approaches that combine RNA-sequencing with proteomics[58,59]. Classical PTM studies, on the other hand, are more commonly based on specific enrichment protocols that enable an in-depth analysis of specific modifications, such as phosphorylations or ubiquitinations, with a focus on peptidoforms rather than proteoforms[3]. Although the COPF strategy can, in principle, detect all types of proteoforms, the current implementation is limited by a minimum peptide set of two. The main reason is that it is difficult to confidently distinguish single outlier peptides from true biological signals. Future work towards improving outlier vs. signal differentiation will therefore further increase the sensitivity of the COPF strategy and the scope of different proteoform groups that can be detected.

Although most presented examples confirm well-annotated proteoforms, our approach has the unique feature of enabling their systematic co-detection and to directly enable the assessment

of their relevance in the studied system, e.g., classifying them as assembly-, cell cycle-, or tissue-specific. The examples discussed in this study only represent a fraction of the wealth of biological information that can be extracted from the COPF analysis of either the SEC-SWATH-MS or the mouse tissue proteomic datasets. To enable researchers to further explore the results in greater depth and to gain an insight into the sensitivity of the approach, we provide an online platform for manual data exploration, which is openly available at http://proteoformviewer.ethz.ch/.

With the constantly increasing number of large-scale datasets and repositories generated by bottom-up proteomics[60–63], there is a wealth of data waiting to be mined for new biology. We envision that our proteoform analysis concept extends previous work[23,25,27] and will thereby contribute to a paradigm shift towards the development of computational methods that directly couple discovery to biological context in such datasets. Strategies with such a direct link will enable easier interpretation of results and selection of promising follow-up candidates.

## Methods

**Proteoform scoring**. The proteoform score is calculated as follows: first, the average peptide Person's correlation is calculated for each cluster. The lowest correlation value across all clusters is selected as within-cluster correlation ($r_{within-cluster}$). Second, the average Pearson's correlation is calculated across all peptides of the protein, referred to as across-cluster correlation ($r_{across-cluster}$). Third, the proteoform score is calculated by subtracting the across-cluster correlation from the within-cluster correlation.

$$\text{Proteoform score} = r_{within-cluster} - r_{across-cluster} \tag{1}$$

A p-value for the proteoform score is derived by first transforming the within- and across-cluster correlations by the Fisher z-transformation:

$$z = \frac{1}{2}\ln\left(\frac{1+r}{1-r}\right) = \text{arctanh}(r) \tag{2}$$

Here, r corresponds to the respective Pearson's correlation. The standard error (SE) of the z-value is defined by:

$$SE = \frac{1}{\sqrt{N-3}} \tag{3}$$

Here, N is the number of samples for which the correlation was calculated. A test statistic, Z, can therefore be derived as follows:

$$Z = \frac{z_{across-cluster} - z_{within-cluster}}{\sqrt{\frac{1}{N_{within-cluster}-3} + \frac{1}{N_{across-cluster}-3}}} \tag{4}$$

The Z-value has an approximately Gaussian distribution under the null hypothesis, assuming that the population correlations are equal. A p-value can therefore be derived as follows:

$$p = 2*(1 - (\text{pnorm}(|(Z)|))) \tag{5}$$

The p-values across all proteins are finally corrected for multiple testing by the BH procedure. The data analyzed in this study was filtered for adj. p-values of 0.1 and a proteoform score of 0.1.

**In silico benchmark**. The in silico benchmark is based on part of the SWATH-MS interlab study[19]. The selected dataset contains 21 replicate HEK293 runs measured on three different days of a week (termed day 1, day 3, and day 5). The data were downloaded from ProteomeXchange (http://proteomecentral.proteomexchange.org) via the PRIDE partner repository[57] with the dataset identifier PXD004886 file site02_global_q_0.01_applied_to_local_global.txt. First, indexed retention time and aqua peptides were removed from the dataset and precursor intensities were summed across charge states. Only peptides without missing values across all 21 runs were considered for further analysis. In addition, proteins with less than four peptides were removed. Median normalization across runs was performed to make intensity values comparable.

To introduce quantitative variation, we generated in silico fold-changes by adjusting intensities of day 3 and day 5 samples with two randomly selected factors between 1 and 6. Subsequently, artificial proteoforms were introduced for 1000 randomly chosen proteins by selecting a specified number of peptides for which the intensity values of day 5 were adjusted by a random factor between 0.01 and 0.9. Depending on the benchmarking setup, either one peptide, two peptides, 50% of the peptides, or a random number between 2 and 50% of peptides per protein were selected.

A script with the complete benchmark analysis is available on GitHub (https://github.com/ibludau/ProteoformAnanlysis/blob/main/PerformanceEvaluation/InterlabBenchmark_final_paper.R).

**COPF analysis of the cell cycle SEC-SWATH-MS dataset**. The peptide-level data and annotation of the cell cycle SEC-SWATH-MS dataset, E1709051521_feature_alignment.tsv and HeLaCCL2_SEC_annotation_full.csv, were downloaded from the Pride repository: https://www.ebi.ac.uk/pride/archive/projects/PXD010288. The data were loaded in R and imported as traces object, using *importMultipleConditionsFromOpenSWATH*. The proteins were further annotated with general information from UniProt using *annotateTraces*. MW calibration of the SEC fractions was performed based on measured standard proteins and their MWs by *calibrateMW* and *annotateMolecularWeight*. Peptide positions within the canonical protein sequence were determined by *annotatePeptideSequences*. Peptides from the same protein with similar start or end position, e.g., generated by missed cleavages, were summarized into a single peptide based on the highest intensity across the dataset by *summarizeAlternativePeptideSequences*. Missing values were imputed for fractions with a valid value in both the preceding and following SEC fraction using *findMissingValues* and *imputeMissingVals*. Peptides detected in fewer than three consecutive fractions across all replicates were excluded from further analysis (*filterConsecutiveIdStretches*). Finally, peptides with zero variance were removed and only proteins with multiple remaining peptides were kept for downstream analysis (*filterSinglePeptideHits*).

For COPF analysis, replicates of each condition were first integrated by *integrateReplicates* followed by appending SEC profiles across conditions by *combineTracesMutiCond*. Subsequently, all pairwise Pearson's correlations between sibling peptides were calculated by *calculateGeneCorrMatrices*. Hierachical clustering based on an average linkage was performed by *clusterPeptides*. The tree was cut into two clusters by *cutClustersInNreal*. Each cluster is required to contain at least two peptides. Peptides that would form a single peptide cluster were marked as outliers. Finally, the proteoform scores and adj. p-values were calculated by *calculateProteoformScore*. Proteoform groups were annotated across all conditions and replicates by *annotateTracesWithProteoforms*.

To evaluate the sequence proximity of the resulting clusters, the *evaluateProteoformLocation* function was applied.

Scripts for the SEC-SWATH-MS COPF analysis are available on GitHub (https://github.com/ibludau/ProteoformAnanlysis/tree/main/CellCycleHela).

**Differential proteoform group analysis of the cell cycle SEC-SWATH-MS dataset**. As a first step of differential proteoform group analysis, protein feature finding was performed. For this, peptide traces were summed across all conditions and replicates by *integrateTraceIntensities*. Peak groups along the SEC dimension were determined by *findProteinFeatures* (corr_cutoff = 0.9, window_size = 7, collapse_method = "apex_only", perturb_cutoff = "5%", rt_height = 1, smoothing_length=7, useRandomDecoyModel = TRUE, quantLevel = "protein_id"). Subsequently, protein features were resolved on the proteoform group level using *resolveProteoformSpecificFeatures* (minProteoformIntensityRatio = 0.1). Features were scored and filtered for a 5% FDR (*scoreFeatures*, FDR = 0.05).

The differential analysis was performed similar as described by Heusel et al.[28] with minor modifications. Intensity values per condition and replicate were first extracted by *extractFeatureVals* and *fillFeatureVals*. The differential analysis was subsequently performed by *testDifferentialExpression*. Tests on the peptide level were finally aggregated to the proteoform group level by *aggregatePeptideTestsToProteoform*. Results were filtered by for a median log2 fold change ≥ 1 and a BH adj. p-value ≤ 0.05.

A script for the differential SEC-SWATH-MS analysis is included on GitHub (https://github.com/ibludau/ProteoformAnanlysis/blob/main/CellCycleHela/06_differentialProteinFeatures_paper.R).

**COPF analysis of the mouse SWATH-MS dataset**. The peptide-level quantitative data from Williams et al.[29] was downloaded and only the whole proteome samples were selected for the analysis with CCprofiler and COPF. The quantitative data were loaded in R and imported as traces object, using *importPCPdata*. The proteins were further annotated with general information from UniProt using *annotateTraces*. Peptide positions within the canonical protein sequence were determined by *annotatePeptideSequences*. Peptides from the same protein with similar start or end position, e.g., generated by missed cleavages, were summarized into a single peptide based on the highest intensity across the dataset by *summarizeAlternativePeptideSequences*. Peptides with zero variance were removed and only proteins with multiple remaining peptides were kept for downstream analysis (*filterSinglePeptideHits*). Due to its suspicious properties, the protein with UniProt accession A2ASS6 was removed.

For COPF analysis, all pairwise Pearson's correlations between sibling peptides were calculated by *calculateGeneCorrMatrices*. Hierachical clustering based on an average linkage was performed by *clusterPeptides*. The tree was cut into two clusters by *cutClustersInNreal*. Each cluster is required to contain at least two peptides. Peptides that would form a single peptide cluster were marked as outliers. Finally, the proteoform scores and adj. p-values were calculated by *calculateProteoformScore*. Proteoform groups were finally annotated by *annotateTracesWithProteoforms*.

To evaluate the sequence proximity of the resulting clusters, the *evaluateProteoformLocation* function was applied.

A script for the mouse tissue data analysis is available on GitHub (https://github.com/ibludau/ProteoformAnanlysis/blob/main/MouseTissue/GetMouseTissueProteoforms_paper.R).

**ANOVA analysis of the mouse SWATH-MS dataset**. For the ANOVA analysis, we selected proteoforms based on a proteoform score cutoff of 0.1 and adj. *p*-value threshold of 0.1. Outlier peptides from the clustering were removed prior to proteoform quantification, which was performed by *proteinQuantification* (quantLevel = "proteoform_id", topN = 1000, keep_less = TRUE). ANOVA analysis for each protein was performed using the *aov* function of the R stats library, applying the following function: log2-intensity ~ tissue * proteoform. Multiple testing across all proteins was performed using a Bonferroni correction (p.adjust of the R stats library).

**Enrichment analyses**. All presented enrichment analyses were performed on the DAVID website[64,65]. Results were filtered for a *p*-value of 0.01, a minimum count of 5, and a minimum fold enrichment of 1.2.

**Sequence proximity analysis**. The sequence proximity analysis evaluates whether peptides assigned to the same proteoform group are in closer relative sequence proximity than expected for random peptide grouping. To test this hypothesis, the peptides of each protein are ranked by their relative peptide position start site (i.e., the position of the first AA of the given peptide in the canonical combined protein sequence). Subsequently, the normalized standard deviation (SD) of the derived peptide position ranks is calculated by dividing the SD of the observed rank vector by the SD of a uniform distribution of equal length. To estimate a probability of whether the observed proximity score is more extreme than expected by chance, we randomly shuffle the peptide ranks 1000 times for each protein and calculate the according proximity score for each of the permutations. An empirical *p*-value for each proteoform group is subsequently derived by dividing the number of random permutations with a more extreme proximity score compared to the real observation by the proximity score of the real observation. For proteins with very few detected peptides, it is impossible to reach statistical significance by means of a classical *p*-value. For these cases, we implemented a second pseudo *p*-value that does not necessitate values "smaller or equal" (as for the classical *p*-value), but that only considers "smaller" values. This means that at a pseudo *p*-value of 10%, maximally 10% of the random permutations score better than the real observation. This criterion covers extreme cases with few peptides, which might still be taken into consideration for follow-up investigations. One example for a protein with two proteoforms that do not reach statistical significance in the sequence proximity analysis, because only two peptides in the short proteoform are too few, is PSMB7 (Supplementary Fig. 2B). You can appreciate that the true observed proximity score is as low as statistically possible, therefore still potentially presenting an interesting candidate, as shown in this example. We report these extreme cases by stating that the protein scored among the lowest 10% of possible *p*-values.

**Phospho-enrichment analysis**. To test our hypothesis that some of the proteoform groups detected by the cell cycle SEC-SWATH-MS analysis might be muti-phosphorylation-derived, we retrieved the cell cycle phospho-proteomic dataset from Karayel et al.[35]. The dataset was downloaded from the original publication Supplementary Table 4 and filtered for 'Mitosis/Interphase' == TRUE and an absolute 'Log2_ratios Mitosis/Interphase' > 0.5. We used the remaining regulated phosphosites to test if proteins with multiple proteoforms in our cell cycle SEC-SWATH-MS dataset are enriched for these sites by performing Fisher's exact test (fisher.test of the R stats library). To further test for each proteoform if more peptides than randomly expected contain a regulated phosphosite (as detected by Karayel et al.[35]), Fisher's exact tests were performed for each proteoform in context of their protein. Proteins with a *p*-value ≤ 0.2 were considered interesting candidates where the proteoform could be related to multiple cell cycle-dependent phosphorylation events.

**Targeted analysis of selected peptides in Skyline**. Selected raw data of both the cell cycle SEC-SWATH-MS and mouse tissue SWATH-MS data were downloaded from their respective Pride repositories https://www.ebi.ac.uk/pride/archive/projects/PXD010288 and https://www.ebi.ac.uk/pride/archive/projects/PXD005044. Targeted extraction of the selected peptides was performed using Skyline version 4.1. The acquisition method was set to DIA and the isolation scheme was matched to the original publication of the datasets. Complete y and b ion series without retention time filtering were extracted for all peptides of interest. The data were than manually inspected for matching peak groups and the fragment ions were filtered for co-elution. The quantification was subsequently based on the total area under the identified peak groups.

**Website**. Data preprocessing and visualization for the dashboard were performed using the python programming language. The following libraries were utilized for data processing: numpy, pandas, scipy, re, and pyteomics[66,67]. Parameter selection, tables, and plots were generated using libraries from the HoloViz family of tools including the following: panel, holoviews, param, bokeh, plotly, and matplotlib. Information about protein domains was retrieved from UniProt (https://www.uniprot.org/, accessed 22 June 2020 for human and 12 July 2020 for mouse),

including following categories: "Chain," "Domain," "Alternative sequence," "Pro-peptide," "Signal peptide," and "Transit peptide."

**PeCorA analysis**. PeCorA analyses were performed as described in the original publication[27]. The threshold_to_filter in *PeCorA_preprocessing* was set to the minimum intensity value. To test the effect of an additional multiple-testing correction on the already adj. *p*-values reported by PeCorA, BH adjustment was performed across all peptides in a dataset (see Supplementary Fig. 1).

A script for the mouse tissue data analysis with PeCorA is available on GitHub (https://github.com/ibludau/ProteoformAnanlysis/blob/main/MouseTissue/MousePecoraAnalysis.R).

**Dynamic proteoform clustering**. Instead of clustering peptides into only two proteoform groups, COPF also offers the possibility to automatically split peptides into any number of proteoform groups by utilizing the function *cutreeDynamic* from the R package dynamicTreeCut[68]. The function is available within COPF as *cutClustersDynamic*. By using a minimum number of two peptides per cluster (min_peptides_per_cluster = 2), single peptide outliers are removed similar to the COPF internal clustering strategy (*cutClustersInN*). Although available in CCprofiler, this strategy has not been extensively tested across multiple datasets and results should be carefully explored.

**Reporting summary**. Further information on research design is available in the Nature Research Reporting Summary linked to this article.

## Data availability

All data presented in this study have been published before and are available on ProteomeXchange (http://proteomecentral.proteomexchange.org) via the PRIDE partner repository[57]. The SEC-SWATH-MS data of HeLa cells in interphase and mitosis were published by Heusel et al.[28] and are available via the identifier PXD010288. The mouse tissue SWATH-MS data were previously published by Williams et al.[29] and are available via the identifier PXD005044. The data used to generate the in silico benchmark were previously published by Collins et al.[19] and are available via the identifier PXD004886.

## Code availability

The CCprofiler workflow including the COPF algorithm is fully implemented and available on GitHub at https://github.com/CCprofiler/CCprofiler and via Zenodo[30]. The code to perform all analyses presented in this study is also available on GitHub at https://github.com/ibludau/ProteoformAnanlysis and via Zenodo[69].

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

## Acknowledgements

We thank Natalie De Souza for reading the manuscript and for providing valuable feedback. We also thank Uwe Schmitt, Patrick Pedrioli, and Pascal Kägi for their help with setting up the website. The project was supported by the SystemsX.ch project PhosphoNetX PPM (to R.A.), the European Research Council (ERC-20140AdG 670821 to R.A.), and the Swiss National Science Foundation project 310030E-173572 awarded under the DACH mechanism to R.A. I.B. was supported by a Swiss National Science Foundation Postdoc.Mobility fellowship (P400PB_191046). G.R. was supported by grants P2EZP3_175127 and P400PB_183933 from the Swiss National Science Foundation. B.C.C. was supported by a Swiss National Science Foundation Ambizione grant (PZ00P3_161435).

## Author contributions

I.B., M.F., M.H., G.R., B.C., H.R., and R.A. conceptualized the project. I.B., M.F., M.H., and H.R. conceptualized the COPF algorithm. I.B., M.F., and Y.C. implemented the COPF algorithm. I.B. and Y.C. performed the benchmarking. I.B. performed the data analysis. C.D. performed the Skyline analysis. I.B. and C.D. interpreted the data with contributions from all authors. I.B. created the website. P.P., B.C., H.R., and R.A. supervised the study. I.B. and R.A. wrote the manuscript with contributions from all authors.

## Competing interests

The authors declare no competing interests.
