## [Peer Review File · Nature Communications]

REVIEWER COMMENTS

Reviewer #1 (Remarks to the Author):

“Systematic detection of functional proteoform groups from bottom-up proteomic datasets” by Bludau et al. describes a method to detect different proteoforms in bottom up or shotgun proteomics datasets. As the authors note, this concept is not new (refs 23, 25-27, and also <https://elifesciences.org/articles/58783>). Their exact implementation called COPF is new. COPF works by computing the pairwise correlation of intensities between all peptides in a protein, clustering those intensity correlations, cutting those clusters into two groups, and then computing a score for the presence of two groups within that protein assignment. They show examples of how this can detect known proteoforms that arise from proteolysis and alternative splicing. They further highlight some examples of indirect detection of PTMs and how COPF suggests new tissue-specific proteoforms.

Overall the conclusions are justified and I can understand how and what they did. The website is super cool and I can appreciate the amount of work that was required to generate it. The figures are beautiful and generally effective. The main issue I'm left with is wondering how this compares to the other options for proteoform detection. How is this new strategy better/worse/complementary?

Major comments:

The main strengths of this work are (1) the web application to explore their results, (2) the application to unique datasets (SEC protein complexes and mouse multi tissue data).

It is unclear from the text whether the peptide profiles for COPF analysis of the SEC data were the average intensity, one fraction, or the profile over SEC fractions. If the latter, then this would be another strength of COPF, which is the ability to apply COPF algorithm to non-pairwise or non-multi-condition comparisons such as the SEC data.

Lines 296-299: “At a false positive rate of 5.6%, 61.7% of the mixed proteins were correctly detected. For 76.8% of these mixed proteins, all peptides were completely and correctly assigned to their parental proteins”. It is hard to know if this is good or bad performance because there is no comparison with other algorithms. The same analysis and statement should be included for the fully synthetic data, and compared with PeCorA <https://github.com/jessemeierlab/PeCorA> and/or another similar competing program(s).

A challenge is that the algorithm forces there to be only 2 peptide groups within a protein. This might not be a safe assumption and is a limitation. The authors do note this in their discussion. Another challenge is that COPF does not directly produce results with p-values but instead an arbitrary cutoff must be determined. Could this problem of determining the number of clusters be re-phrased as a statistical test somehow? Could this problem be solved along with the problem of forcing 2 clusters with statistical tests asking whether there are 1, 2, 3, or more clusters? For example, <https://www.ncbi.nlm.nih.gov/pmc/articles/PMC3184008/> ?

The COPF cutoff used for ROC analysis in Figure 2F should be provided in the legend and text, along with how it was determined at least in the text.

More generally, how is the COPF protein isoform score cutoff chosen for each different experiment? Maybe I missed it, but this should be clearly in the results when describing that scores distributions are plotted near figure 1.

Please update citation #26 to the accepted version <https://doi.org/10.1021/acs.jproteome.0c00602>

Figure 6: Again, how does this compare with other proteoform detection algorithms such as PeCorA or FlexiQuant-LF? How is it the same, different, and complementary?

Minor comments:

The 3-paragraph abstract is unconventional and I find it difficult. I suggest compressing to ~150 words and moving other ideas to the intro if they are not already there.

I suggest removing figure 2A-2C because I don't think it is as good as the synthetic data and I find it confusing to your message. The assumption that all those proteins should be single groups is probably not true and muddies the results. While reading it I was imagining a better synthetic test case, which was then introduced after this.

The example in figure 5A/B fits better with known examples in figure 4.

Line 593: "observed at elution ~42kDa +/- 10 kDa", if the authors want to state this they must provide how it was determined.

The discussion is very long and therefore difficult to read. Much of it repeats the introduction. I suggest shortening it significantly to focus on adding value rather than repeating the same ideas. E.g. Do you need the full 1st, 3rd, 4th paragraphs?

Reviewed by Jesse G. Meyer

Reviewer #2 (Remarks to the Author):

The manuscript, “Systematic detection of functional proteoform groups from bottom-up proteomic datasets” is a well-written, well-executed demonstration of proteoform inference from bottom-up proteomics data. This is an important area, as while the comprehensive analysis of proteoforms in complex systems is increasingly recognized as being important, top-down proteomics tools for accomplishing that provide substantially less proteome coverage than do the more widely used bottom-up strategies. Although as recognized by the authors, bottom-up is fundamentally unable to definitively identify or quantify proteoforms due to the loss of protein context engendered by the protease digestion step, there is nonetheless a great deal of valuable and pertinent information that can be gleaned from bottom-up data. This report described an advance of that type. The authors present a four-step, correlation-based algorithm for the prediction of proteoforms and include in-silico benchmarking, a discussion of parameters affecting the scoring of potential proteoforms, and a demonstration of detection of proteoforms in two complex samples. Using their method, the authors were successfully able to detect and contextualize proteoforms, including alternative splicing events, post-translational modifications, and post-translational proteolytic processing. The citations provide excellent documentation of the advances within their field and properly places the manuscript with respect to the development of methods inferring proteoforms from bottom-up experiments. We recommend this manuscript be accepted for publication subject to addressing the following minor revisions:

- The third step in the algorithm clusters the proteins based on one minus the pairwise correlation (dissimilarity correlation) between peptides, and the fourth step assigns a proteoform score. It is not clear from the text whether the fourth step uses correlation or dissimilarity correlation for the score assignment.
- In Figure 6A, quadriceps is spelled incorrectly in the caption of the arrow pointing at protein A.

Reviewer #3 (Remarks to the Author):

- What are the noteworthy results?

This paper presents a “new” approach to Correlation based function ProteoForm assessment (COPF), which is indeed an important area in proteomics. The noteworthy results are the interesting results

on real datasets in the context of potential proteoforms and what they mean. However, it does not show if existing approaches would have found the same results.

- Will the work be of significance to the field and related fields? How does it compare to the established literature? If the work is not original, please provide relevant references.

The primary issue with this manuscript in its current form is a lack of comparison to existing methods. The paper by Forshed (Reference #25) is essentially a clustering based strategy to assign peptides in exactly the manner that is defined by the authors: "An important distinction between functional proteoform groups assigned by COPF and those determined by top-down proteomics approaches is that COPF does not fully characterize the proteoform's complete primary amino acid sequence and all of its modifications. It merely determines whether peptides exist that can differentiate the different biological contexts of a protein."

From the abstract of Forshed paper: "The method is based on the assumption that the quantitative pattern of peptides derived from one protein will correlate over several samples. Dissonant patterns arise either from outlier peptides or because of the presence of different protein species. By correlation analysis, protein quantification by peptide quality control identifies and excludes outliers and detects the existence of different protein species. Alternative protein species are then quantified separately." This sounds identical to the approach by COPF. The Forshed paper even identifies outliers in the same way as noted by this paper.

Further the work by Webb-Robertson takes an alternate approach using statistics-derived patterns, but does exactly the same thing and compares to the PQQP approach of Forshed. The paper does not outline how this approach is original from the PQQP approach, although they do cite the prior work. It appears to be a better R package perhaps using the same approach? The difference should be explicitly defined.

- Does the work support the conclusions and claims, or is additional evidence needed?

As mentioned, the lack of a comparison with existing approaches (References 23, 24 and 25) doesn't support a claim "We envision that our proteoform analysis concept will contribute to a paradigm shift towards the development of computational methods that directly couple discovery to biological context in such datasets." The PQQP paper started this conversation in 2011 with a correlation-based clustering approach for peptide. The results from the datasets are interesting, but do not support that this is a novel approach to systematically assign peptides to covarying proteoforms. The code available for both PQQP and BP-Quant and likely others also offer a pipeline to do this.

- Is the methodology sound? Does the work meet the expected standards in your field?
- Is there enough detail provided in the methods for the work to be reproduced?

The actual method is mostly described in just Figure 1. It is noted that different thresholds for correlation are used in different analyses. In the methods each dataset analysis is described as a series of R functions that are put together in a workflow. A more clear description in writing should be given in addition to Figure 1 with how each function maps to each step in the process. It also would be good to specify where users need to set specific thresholds.

Point-by-point response to reviewer comments

Reviewer #1 (Remarks to the Author):

“Systematic detection of functional proteoform groups from bottom-up proteomic datasets” by Bludau et al. describes a method to detect different proteoforms in bottom up or shotgun proteomics datasets. As the authors note, this concept is not new (refs 23, 25-27, and also <https://elifesciences.org/articles/58783>). Their exact implementation called COPF is new. COPF works by computing the pairwise correlation of intensities between all peptides in a protein, clustering those intensity correlations, cutting those clusters into two groups, and then computing a score for the presence of two groups within that protein assignment. They show examples of how this can detect known proteoforms that arise from proteolysis and alternative splicing. They further highlight some examples of indirect detection of PTMs and how COPF suggests new tissue-specific proteoforms.

Overall the conclusions are justified and I can understand how and what they did. The website is super cool and I can appreciate the amount of work that was required to generate it. The figures are beautiful and generally effective. The main issue I’m left with is wondering how this compares to the other options for proteoform detection. How is this new strategy better/worse/complementary?

Major comments:

The main strengths of this work are (1) the web application to explore their results, (2) the application to unique datasets (SEC protein complexes and mouse multi tissue data).

It is unclear from the text whether the peptide profiles for COPF analysis of the SEC data were the average intensity, one fraction, or the profile over SEC fractions. If the latter, then this would be another strength of COPF, which is the ability to apply COPF algorithm to non-pairwise or non-multi-condition comparisons such as the SEC data.

The COPF analysis of the SEC data is performed across the entire SEC profiles including intensity information of all fractions. The strategy, therefore, enables the possibility to apply the algorithm to a single sample e.g. with the goal to detect assembly-specific proteoforms within a single experimental condition but also to perform comparisons across conditions. The ability of COPF to exploit inherent variation in the sample (introduced through SEC, genetic differences in subjects or controlled experiments) is indeed one of the strengths of COPF. Thus, its capacity to detect proteoforms in a single condition is a major advantage of the COPF approach compared to previous software tools and we thank the reviewer for pointing out that this feature was not stated clearly enough in the original manuscript. We

have clarified the SEC analysis steps and highlighted specific application cases and benefits of COPF in the revised manuscript text.

Lines 411-413:

“Importantly, intensities of individual peptides across all measured fractions and conditions were considered for calculating the pairwise correlations in COPF and to derive the scoring metrics.”

Lines 226-237:

“It is important to highlight that COPF analysis does not require any prior definition of biological conditions or a specific experimental design, since it exploits inherent variation in the data independent of its origin. Thus, COPF has the unique benefit that it is applicable to non-pairwise comparisons or to comparisons that do not include multiple conditions, exemplified by continuous data such as a single SEC-SWATH-MS experiment where a single condition is analyzed. Additionally, COPF can also be applied to data with complex, nested designs including multiple covariates. In fact, the correlation-based approach employed by COPF is particularly powerful when applied to large and heterogeneous datasets, even in the absence of an explicit reference condition. These are typically difficult to assess by other approaches, such as PeCorA 27, that require a homogeneous reference condition. “

Lines 296-299: “At a false positive rate of 5.6%, 61.7% of the mixed proteins were correctly detected. For 76.8% of these mixed proteins, all peptides were completely and correctly assigned to their parental proteins”. It is hard to know if this is good or bad performance because there is no comparison with other algorithms. The same analysis and statement should be included for the fully synthetic data, and compared with PeCorA <https://github.com/jessegmeyerlab/PeCorA> and/or another similar competing program(s).

This is an important point and we have carried out a significant amount of new analyses and accordingly revised and extended the manuscript to address this comment. Most importantly, we now include a comprehensive benchmark to compare COPF with PeCorA. Due to fundamental differences in the design of both software tools, both previous benchmarking sets were not applicable for the comparison:

- 1) PeCorA could not be applied to our original benchmarking data because it was based on a single condition SEC-SWATH-MS dataset. PeCorA requires two or more predefined conditions to work.
- 2) A comparison based on the second, fully simulated dataset was also not possible, because in contrast to COPF, which works purely based on correlation, PeCorA takes actual peptide intensities into account. The simulated dataset was not designed for this type of analysis.

Thus, to perform a meaningful comparative benchmark, we generated a new benchmarking dataset that is compatible with both PeCorA and COPF. The basis for the new benchmarking

data is a subset of the SWATH-MS interlab study (Collins et al., 2017) consisting of 21 replicate HeLa samples (7 replicates measured on day 1, day 3 and day 5 each within a week). To introduce quantitative variation, we generated *in silico* fold-changes by adjusting intensities of day 3 and day 5 samples with two randomly selected factors between 1 and 6. Subsequently, artificial proteoforms were introduced for 1000 proteins by selecting a specified number of peptides for which the intensity values of day 5 were adjusted by a random factor between 0.01 and 0.9. The new benchmark is described in detail in the revised manuscript lines 395-364, Figure 2 and Supplementary Figure 1.

In brief, the benchmark results indicate complementary use cases for both PeCorA and COPF. While PeCorA can detect proteoforms using single peptides as evidence, COPF was specifically designed for detecting proteoform groups using the signals of multiple peptides. This difference has important implications on the interpretation of results obtained by PeCorA or COPF. While PeCorA can detect single outlier peptides with high sensitivity, it is difficult to distinguish whether the detected signal is caused by a technical outlier or by a true biological signal. Proteoforms reported by COPF require at least two peptides differing between proteoforms, thus reducing the chance of observing a purely technical artefact. Additionally, the grouping of peptides by COPF enables the direct report of proteoform groups. PeCorA only reports single peptides differing in intensity between samples but does not derive conclusions on whether multiple outlier peptides of the same protein co-vary because they are derived from the same proteoform or from different proteoforms. The type of proteoforms detected by PeCorA and COPF therefore differ. While PeCorA is sensitive in detecting single peptide proteoforms, such as caused by single PTMs, COPF is more suitable for proteoforms involving longer sequence stretches, such as proteoforms derived by proteolytic cleavage or alternative splicing and also multiple co-regulated PTMs.

Our benchmarking results further indicate that the adjusted p-values from COPF are better calibrated than those from PeCorA on our dataset and we conclude that PeCorA p-values are not directly applicable to derive an FDR on the proteome-wide level and are likely better interpreted in a more limited probabilistic sense (Figure 2D and Supplementary Figure S1B). Proteoform numbers reported by PeCorA and COPF at the same specified FDR are therefore not directly comparable. We therefore used ROC curves over a wide range of adjusted p-value thresholds instead of a single adjusted p-value threshold to show sensitivity versus specificity of PeCorA and COPF as comparative metric. This analysis shows comparable performance of the two methods with an advantage of PeCorA for single peptide isoforms and an advantage of COPF for multi-peptide isoforms.

A challenge is that the algorithm forces there to be only 2 peptide groups within a protein. This might not be a safe assumption and is a limitation. The authors do note this in their discussion. Another challenge is that COPF does not directly produce results with p-values but instead an arbitrary cutoff must be determined. Could this problem of determining the

number of clusters be re-phrased as a statistical test somehow? Could this problem be solved along with the problem of forcing 2 clusters with statistical tests asking whether there are 1, 2, 3, or more clusters? For example, <https://www.ncbi.nlm.nih.gov/pmc/articles/PMC3184008/> ?

We agree with the reviewer that these are important points to consider and we have substantially revised the manuscript to address them. Unfortunately, the method and paper suggested by the reviewer was not applicable to the present situation because of (1) the availability of only a limited number of peptides per protein for clustering and (2) our desire to exclude single outlier peptides from driving the clustering.

We nevertheless understand the reviewers concern and have extended the COPF approach to (1) enable the detection of more than two proteoforms per protein and (2) to derive a strategy for estimating p-values and an FDR for the derived proteoforms. Regarding point 1, we integrated a new clustering approach in COPF that automatically determines the number of clusters, while still maintaining the strategy of requiring minimally two peptides per cluster (Dynamic Tree Cut library in R). Although more than two proteoforms could be detected by this new approach, our analyses showed that only 1 protein in the SEC and 2 proteins in the mouse data were determined to have 3 proteoforms (data not shown). It is important to keep in mind that these results are dependent on the parameter adjustments of the clustering approach and users are able to adjust the clustering settings to evaluate results of more fine-grained proteoform evaluation. Due to the minor effect of the new clustering analyses in the datasets tested, we kept the initial clustering approach. However, we now offer the multi-proteoform cluster option as parameter in the COPF analysis. We describe the availability of the more fine-grained clustering in the discussion (lines: 844-854) and the method section (lines: 1177-1186).

Regarding the scoring threshold selection (point 2), we have made substantial progress by including a new p-value estimation strategy (see methods section lines 933-967). Figure 2D and Supplementary Figure 1B demonstrate that the new statistical approach provides accurate, yet conservative FDR estimates in the updated benchmarking dataset. We have subsequently performed the p-value estimation on the datasets analyzed in this study and decided to apply a 2-way cutoff, analogous to standard T-test analyses, by requiring a 10% adjusted p-value and a minimal proteoform score of 0.1 (which corresponds to a minimal difference of within versus across cluster correlation of 0.1 in Pearson correlation space). This satisfies a minimal effect size criterion and precludes that very small differences in correlation (below 0.1) are reported.

All results presented in the manuscript have been updated according to the revised workflow. Our analysis now reports 317 proteins with proteoforms in the SEC-SWATH-MS dataset and

63 proteins with proteoforms in the mouse tissue data at 10% FDR and a minimal proteoform score of 0.1.

The reviewer's comments, therefore, guided us to significantly improve the software and we hope that this alleviates the reviewer's concerns. Nevertheless, we still encourage users to visually explore the results on the website and to critically judge their own experimental results.

The COPF cutoff used for ROC analysis in Figure 2F should be provided in the legend and text, along with how it was determined at least in the text.

We thank the reviewer for pointing this out. Since the initial benchmarking data are no longer part of the revised manuscript version, this Figure is no longer used. However, the legends for ROC curves in the new benchmarking figure now clearly state that each point corresponds to the results of a specific adjusted p-value cutoff, i.e. curves were generated by iterating over adjusted p-values between 0 and 1.

Main text lines: 319-322

"We compared the receiver operating characteristic (ROC) curves for COPF and PeCorA for all three benchmarking sets (Figure 2C). Here, individual points are derived by filtering the data at a specific adjusted p-value threshold and showing the corresponding true positive rate (TPR) and false positive rate (FPR)."

Figure legend lines: 375-378

"Receiver operator characteristic (ROC) curves for three in silico benchmark datasets. Individual points in the curve are generated by iterating over different adjusted p-value thresholds. The datapoints derived from an adjusted p-value threshold of 0.1 are highlighted by a red circle."

More generally, how is the COPF protein isoform score cutoff chosen for each different experiment? Maybe I missed it, but this should be clearly in the results when describing that scores distributions are plotted near figure 1.

We hope that the newly included p-value estimation in COPF addresses the reviewer's concern. Now we applied a 10% FDR threshold in addition to a proteoform score threshold of 0.1. This can be regarded similar to a 2-sided threshold frequently applied to T-test results where both a significance and a fold-change cutoff are selected.

Please update citation #26 to the accepted version
<https://doi.org/10.1021/acs.jproteome.0c00602>

We thank the reviewer for pointing us to the accepted manuscript and we have updated the reference.

Figure 6: Again, how does this compare with other proteoform detection algorithms such as PeCorA or FlexiQuant-LF? How is it the same, different, and complementary?

We have performed the analysis of the mouse tissue dataset with PeCorA. Results are shown in Figure 6E and described in the main text:

lines 665-674:

“Finally, we compared the proteoform containing proteins called by COPF with those determined by PeCorA 27 (Figure 6E). In total, PeCorA reported significant peptides for 2730 out of 2885 proteins (95%) at an adjusted p-value cutoff of 10%. This set covers all but one proteoform containing protein determined by COPF. If an additional multiple testing correction of the adjusted p-values reported by PeCorA was performed across all peptides, the number of proteins reported by PeCorA dropped only by 41, still covering 93% of all proteins. These findings are in line with the benchmarking results (see above), suggesting that PeCorA is more sensitive than COPF at determining outlier peptides, however at the cost of a potentially high false positive rate.”

We think that the results demonstrate the high sensitivity of PeCorA. However, we assume that a large proportion of the reported findings by PeCorA result from non-biological effects that could also be explained by technical artefacts based on single peptide measurements in LC-MS/MS. In contrast to the original application datasets for which PeCorA was designed and shown to perform reliably, both the benchmarking and mouse tissue study represent datasets with more replicates and higher between sample variability. It is known from other proteomics studies that both increasing numbers of samples and higher variability between samples have an impact on the number of reported false positives and they need to be more carefully controlled in such increasingly complex datasets.

In such a setting, COPF is likely very conservative, without the possibility to find any single-peptide effects and thus lowering recall. By only reporting proteoforms with two or more co-regulated peptides, the likelihood of technical artefacts driving the effect is markedly reduced and the proteoform groups by COPF are likely to be functionally relevant. Nevertheless, we expect the real number of biologically relevant proteoforms in the dataset to lie somewhere in between what COPF and PeCorA report.

FlexiQuant-LF was designed to run on single proteins and does not provide a practical approach for large, systems biology studies like the ones presented herein. We therefore only performed the comparison with PeCorA.

Minor comments:

The 3-paragraph abstract is unconventional and I find it difficult. I suggest compressing to ~150 words and moving other ideas to the intro if they are not already there.

We have shortened the abstract and formatted the text as a single paragraph.

I suggest removing figure 2A-2C because I don't think it is as good as the synthetic data and I find it confusing to your message. The assumption that all those proteins should be single groups is probably not true and muddies the results. While reading it I was imagining a better synthetic test case, which was then introduced after this.

Since we now performed a completely new benchmark, we excluded the two previous datasets to solely focus the reader on the relevant comparisons.

The example in figure 5A/B fits better with known examples in figure 4.

We agree with the reviewer regarding Figure content. However, we think Figure 4 is already very information dense and the content is better digestible when keeping Figure 4 and 5 as they are.

Line 593: "observed at elution ~42kDa +/- 10 kDa", if the authors want to state this they must provide how it was determined.

The approximate molecular weight of each SEC fraction can be estimated based on an external standard set of reference proteins with known MWs fractionated on the same SEC setup. *CCprofiler* can thus establish a transformation function based on the log-linear relationship between elution fractions and apparent MWs inherent to SEC, thus enabling the annotation of all sampled fractions with an apparent MW. We added an explanatory statement in the manuscript lines: 565-567.

The discussion is very long and therefore difficult to read. Much of it repeats the introduction. I suggest shortening it significantly to focus on adding value rather than repeating the same ideas. E.g. Do you need the full 1st, 3rd, 4th paragraphs?

We have substantially shortened the discussion and sharpened the message. The 1st paragraph was removed and the 3rd and 4th paragraph significantly shortened and combined.

Reviewed by Jesse G. Meyer

Reviewer #2 (Remarks to the Author):

The manuscript, "Systematic detection of functional proteoform groups from bottom-up proteomic datasets" is a well-written, well-executed demonstration of proteoform inference from bottom-up proteomics data. This is an important area, as while the comprehensive analysis of proteoforms in complex systems is increasingly recognized as being important, top-down proteomics tools for accomplishing that provide substantially less proteome coverage than do the more widely used bottom-up strategies. Although as recognized by the authors, bottom-up is fundamentally unable to definitively identify or quantify proteoforms due to the loss of protein context engendered by the protease digestion step, there is nonetheless a great deal of valuable and pertinent information that can be gleaned from bottom-up data. This report described an advance of that type. The authors present a four-step, correlation-based algorithm for the prediction of proteoforms and include in-silico benchmarking,

a discussion of parameters affecting the scoring of potential proteoforms, and a demonstration of detection of proteoforms in two complex samples. Using their method, the authors were successfully able to detect and contextualize proteoforms, including alternative splicing events, post-translational modifications, and post-translational proteolytic processing. The citations provide excellent documentation of the advances within their field and properly places the manuscript with respect to the development of methods inferring proteoforms from bottom-up experiments.

We recommend this manuscript be accepted for publication subject to addressing the following minor revisions:

- The third step in the algorithm clusters the proteins based on one minus the pairwise correlation (dissimilarity correlation) between peptides, and the fourth step assigns a proteoform score. It is not clear from the text whether the fourth step uses correlation or dissimilarity correlation for the score assignment.

In contrast to the clustering itself, the final scores are derived from correlation values (not dissimilarity). We have clarified the documentation of the fourth step in both the figure legend (lines 277-278) and added more detailed description of the proteoform score calculation in the method section lines: 933-967.

In Figure 6A, quadriceps is spelled incorrectly in the caption of the arrow pointing at protein A.

We thank the reviewer for pointing out this error and have corrected the typo in the revised manuscript.

Reviewer #3 (Remarks to the Author):

- What are the noteworthy results?

This paper presents a “new” approach to Correlation based function ProteoForm assessment (COPF), which is indeed an important area in proteomics. The noteworthy results are the interesting results on real datasets in the context of potential proteoforms and what they mean. However, it does not show if existing approaches would have found the same results.

To address the important issue stated by the reviewer we have substantially revised the manuscript, which now contains a benchmark comparison to the recently published tool PeCorA. Please also see comments to reviewer 1, manuscript lines 295-364 as well as Figures 2 and 6E.

- Will the work be of significance to the field and related fields? How does it compare to the established literature? If the work is not original, please provide relevant references.

The primary issue with this manuscript in its current form is a lack of comparison to existing methods. The paper by Forshed (Reference #25) is essentially a clustering based strategy to assign peptides in exactly the manner that is defined by the authors: “An important distinction between functional proteoform groups assigned by COPF and those determined by top-down proteomics approaches is that COPF does not fully characterize the proteoform’s complete primary amino acid sequence and all of its modifications. It merely determines whether peptides exist that can differentiate the different biological contexts of a protein.”

From the abstract of Forshed paper: “The method is based on the assumption that the quantitative pattern of peptides derived from one protein will correlate over several samples. Dissonant patterns arise either from outlier peptides or because of the presence of different protein species. By correlation analysis, protein quantification by peptide quality control identifies and excludes outliers and detects the existence of different protein species. Alternative protein species are then quantified separately.” This sounds identical to the approach by COPF. The Forshed paper even identifies outliers in the same way as noted by this paper.

Further the work by Webb-Robertson takes an alternate approach using statistics-derived patterns, but does exactly the same thing and compares to the PQPQ approach of Forshed. The paper does not outline how this approach is original from the PQPQ approach, although they do cite the prior work. It appears to be a better R package perhaps using the same approach? The difference should be explicitly defined.

In the initial submission, we purposefully refrained from a comparative analysis, for two main reasons, (1) PQPQ, which is (as the reviewer correctly points out) most similar to our approach, is no longer available (also not upon request to the corresponding author) and (b) the few other tools, e.g. PeCorA or BP-Quant, were optimized for very different goals and are

applicable to different types of datasets, in the case of PeCorA and BP-Quant for datasets with pre-defined experimental conditions and not to continuous data like COPF (or PQPQ). Additionally, these other tools are focused on the identification of single outlier peptides instead of entire co-regulated proteoform groups. COPF thereby provides a more conservative procedure for proteoform detection, which restricts reported results to those proteoform groups that are likely of biological relevance. However, to address the reviewers concern, we have now substantially revised the manuscript. We now included a benchmark against the most recently published tool PeCorA. Please also see comments to reviewer 1, manuscript lines 295-364 as well as Figures 2 and 6E.

As stated above, we could not compare COPF to PQPQ, because the tool is no longer available (also not upon request to the corresponding author). Although we could not perform the direct comparison with PQPQ, we would like to highlight the differences between the tools and the specific benefits of the COPF workflow:

- 1) After our revisions, COPF includes a statistical model to estimate an FDR for proteoform detection that can directly be used to filter the data at a desired FDR threshold.
- 2) COPF includes post-processing options such as the peptide proximity analysis. Here proteoforms get annotated with additional information about their characteristics. Proteoforms with a high sequence proximity are for example likely to be derived from alternative splicing or proteolytic cleavage.
- 3) COPF is directly integrated in the CCProfiler library for protein complex analysis in CoFrac-MS data. It is therefore directly possible to assess assembly characteristics of the determined proteoforms. Together with point 2, these are unique features of our workflow which provide biologically meaningful insights to the discovered proteoforms that were not directly extractable from previous workflows such as results derived from PQPQ.

Comments to other tools that infer proteoforms:

- 1) FlexiQuant-LF: This tool was designed to run on single proteins. A systems biology application to proteome-wide data as presented herein is therefore currently not feasible.
- 2) BP-Quant (Webb-Robertson et al.): This tool was designed to operate on data containing pre-defined conditions and non-continuous data. Similar to PeCorA it is therefore not directly suitable for CoFrac-MS data analysis. To still enable a performance comparison between software that is aware of the experimental design, we decided to compare COPF to the state-of-the-art PeCorA tool instead of BP-Quant.

- Does the work support the conclusions and claims, or is additional evidence needed?

As mentioned, the lack of a comparison with existing approaches (References 23, 24 and 25) doesn't support a claim "We envision that our proteoform analysis concept will contribute to a paradigm shift towards the development of computational methods that directly couple discovery to biological context in such datasets." The PQQP paper started this conversation in 2011 with a correlation-based clustering approach for peptide. The results from the datasets are interesting, but do not support that this is anovel approach to systematically assign peptides to covarying proteoforms. The code available for both PQQP and BP-Quant and likely others also offer a pipeline to do this.

We agree and rephrased our statements in the revised manuscript:

lines: 901-904

"We envision that our proteoform analysis concept extends previous work^{23,25,27} and will thereby contribute to a paradigm shift towards the development of computational methods that directly couple discovery to biological context in such datasets."

We would further like to point out that a main advance of our method is the direct integration of the reported proteoforms with strategies for their biological characterization by means of proximity analysis and, for CoFrac-MS data, with protein complex assembly analysis within the CCprofiler framework. This is of great importance for experimental biology because there are numerous techniques that have been developed that identify chemical entities, including metabolites, nucleic acids and polypeptides at an increasing pace but the strategies to determine whether or not these entities are functionally relevant have been lagging. The COPF algorithm addresses this issue because there is a high likelihood that two proteoforms that associate with different complexes contribute to different cellular functions. We have revised the manuscript text to more clearly highlight the main application area of COPF for complex experimental designs, and especially SEC-SWATH-MS data.

- Is the methodology sound? Does the work meet the expected standards in your field?
- Is there enough detail provided in the methods for the work to be reproduced?

The actual method is mostly described in just Figure 1. It is noted that different thresholds for correlation are used in different analyses. In the methods each dataset analysis is described as a series of R functions that are put together in a workflow. A more clear description in writing should be given in addition to Figure 1 with how each function maps to each step in the process. It also would be good to specify where users need to set specific thresholds.

We have significantly revised the COPF workflow and its description to now include an FDR estimation step. This enables the user to select the same cutoff for all analyses. Here we decided to go for a 10% FDR in combination with a minimum proteoform score of 0.1 in all datasets. We have further expanded on the technical details of the scoring in the method section lines: 933-967.

REVIEWERS' COMMENTS

Reviewer #1 (Remarks to the Author):

Excellent work; the authors went above and beyond my suggestions.

-Jesse Meyer

Reviewer #3 (Remarks to the Author):

Overall this is an excellent revision and has addressed my largest concerns. Figure 1 is significantly improved and the method is much easier to follow. I also missed the first time the proximity analysis, which does make this method unique. "The proximity analysis evaluates if peptides assigned to the same proteoform group are in closer relative sequence proximity than expected for random peptide grouping."

Minor Issues:

I feel like the way that the results in Figure 2 are presented via ROC analysis are confusing. The article says:

"The first consisted of proteins where proteoforms differed by a single peptide" and then

"Thus, COPF did not, as expected, find proteoforms in the first benchmarking set (Figure 2C, left panel)." This is confusing. I feel like the method should be able to say there is no proteoforms (i.e., 1 protein form versus >1 protein form) identified and so it gets it right so it should show as a TPR in the ROC curve. How it is presented it looks like you get no information if there aren't proteoforms. It is a minor presentation and interpretation component of how the algorithm works, but I think it would be useful to rethink how this is presented.

Point-by-point response to reviewer comments

Reviewer #3 (Remarks to the Author):

Overall this is an excellent revision and has addressed my largest concerns. Figure 1 is significantly improved and the method is much easier to follow. I also missed the first time the proximity analysis, which does make this method unique. “The proximity analysis evaluates if peptides assigned to the same proteoform group are in closer relative sequence proximity than expected for random peptide grouping.”

Minor Issues:

I feel like the way that the results in Figure 2 are presented via ROC analysis are confusing. The article says: “The first consisted of proteins where proteoforms differed by a single peptide” and then “Thus, COPF did not, as expected, find proteoforms in the first benchmarking set (Figure 2C, left panel).” This is confusing. I feel like the method should be able to say there is no proteoforms (i.e., 1 protein form versus >1 protein form) identified and so it gets it right so it should show as a TPR in the ROC curve. How it is presented it looks like you get no information if there aren't proteoforms. It is a minor presentation and interpretation component of how the algorithm works, but I think it would be useful to rethink how this is presented.

We are delighted that our revisions addressed the reviewer's previous concerns. We also thank the reviewer for pointing out that the presentation of the new benchmarking results in Figure 2C could be misleading. The ROC curve for the first benchmarking dataset (Figure 2C, left panel) indeed shows that COPF is incapable to determine that a protein has multiple proteoform groups if the proteoform differ by a single detected peptide. This is expected behavior of COPF based on its design. In contrast, PeCorA was specifically designed to detect proteoforms differing by single peptides and that is why it performs very well in the first benchmarking set. We revised the text in the manuscript to state more clearly what results we would expect and how the ROC curves agree with expectations.

Lines 225 - 234:

“COPF requires minimally two peptides to differentiate proteoform groups. As expected, COPF could not detect the proteoforms differing by a single peptide in the first benchmarking set (Figure 2C, left panel). In contrast, PeCorA was specifically designed for detecting proteoforms differing by a single peptide and could, therefore, achieve a convincing ROC curve. In the second benchmarking set proteoform groups differed by two peptides and COPF and PeCorA show similar ROC curves (Figure 2, middle panel, and Supplementary Figure S1A). Here, COPF has slightly higher TPRs in the lower range of FPRs between 0 and 0.1. Finally, COPF markedly outperformed PeCorA in the third benchmarking set in which proteoform groups differed by 50% of the protein's peptides (Figure 2C, right panel).”